# `AdLift`: Lifting Adversarial Perturbations to Safeguard 3D Gaussian Splatting Assets Against Instruction-Driven Editing

Ziming Hong[1]   Tianyu Huang[1]   Runnan Chen[1]   Shanshan Ye[2]   Mingming Gong[3,2]   Bo Han[4]   Tongliang Liu[1,2]

## Abstract

Recent studies have extended instruction-driven 2D editing pipelines to 3D Gaussian Splatting (3DGS), enabling faithful 3DGS asset manipulation for advanced content creation. However, it also exposes 3DGS assets to serious risks of unauthorized editing and malicious tampering. Although adversarial perturbations against editing models have proven effective for protecting 2D images, applying them to 3DGS encounters two major challenges: *view-generalizable protection* and *balancing invisibility with protection capability*. In this work, we propose `AdLift`, a novel editing safeguard for 3DGS that prevents instruction-driven editing across arbitrary views and dimensions by lifting strictly bounded 2D adversarial perturbations into 3D Gaussian-represented safeguard. To ensure both *protective effectiveness* and *invisibility*, these safeguard Gaussians are progressively optimized across training views using a tailored Lifted PGD, which first conducts *gradient truncation* during backpropagation from the editing model to the rendered image and applies projected gradient updates to strictly bound image-level perturbations. Then, the resulting perturbation is backpropagated to the safeguard Gaussian parameters via *image-to-Gaussian fitting*. We alternate these two steps, yielding effective and imperceptible protection that generalizes across both training and novel views. Empirically, qualitative and quantitative results demonstrate that the proposed `AdLift` effectively protects against state-of-the-art instruction-driven 2D and 3DGS editing.

## 1. Introduction

3D representation has become a key topic in computer vision and graphics, powering applications such as film production, game development, virtual reality, and autonomous driving. Among recent advances, 3D Gaussian Splatting (3DGS) (Kerbl et al., 2023) has emerged as a transformative approach that combines photorealistic fidelity with real-time rendering efficiency through an explicit Gaussian-based representation. This enables 3DGS to rapidly integrate into pipelines for creating, sharing, and deploying 3D assets across diverse downstream applications, where the resulting assets naturally carry substantial value as digital content.

Recently, text-to-image latent diffusion models (LDMs) have enabled efficient and high-quality 2D image editing guided by human instructions (Brooks et al., 2023). Building upon these advances, a growing body of work has extended instruction-driven 2D editing pipelines to 3DGS (Vachha & Haque, 2024; Chen et al., 2024b). While these methods substantially facilitate 3DGS content creation, they also expose 3DGS assets to serious risks of unauthorized modification and malicious tampering. As shown in Figure 1(a), anyone with access to 3DGS assets can perform arbitrary instruction-based editing, even without permission. *Such risks raise urgent concerns about the intellectual property (IP) (Li et al., 2025b; Zhao et al., 2026) of 3DGS assets.* Although a few prior works have explored IP protection for 3DGS, they mainly focus on *passive protection techniques* such as 3DGS watermark and steganography (Huang et al., 2024; Zhang et al., 2025). These passive approaches, which are only applicable for verifying asset ownership after release, are incapable of defending against instruction-driven malicious editing enabled by latent diffusion models such as InstructPix2Pix (Brooks et al., 2023).

As such, this raises a natural question: *can we proactively protect 3DGS assets against malicious and unauthorized editing?* Since existing 3DGS editing pipelines build upon 2D editing, and several works have proposed protecting 2D images by launching *adversarial attacks* against LDMs (Liang et al., 2023; Xue et al., 2024; Wang et al., 2025), it is natural to ask whether these 2D protection methods can be directly extended to 3DGS. The answer is **no**. Two key challenges arise in achieving effective protection for 3DGS:

[1]Sydney AI Centre, The University of Sydney [2]Mohamed bin Zayed University of Artificial Intelligence [3]The University of Melbourne [4]Hong Kong Baptist University. Correspondence to: Shanshan Ye <shanshan.ye@mbzuai.ac.ae>. Code: https://github.com/tmllab/2026_ICML_AdLift.

*Proceedings of the $43^{rd}$ International Conference on Machine Learning*, Seoul, South Korea. PMLR 306, 2026. Copyright 2026 by the author(s).

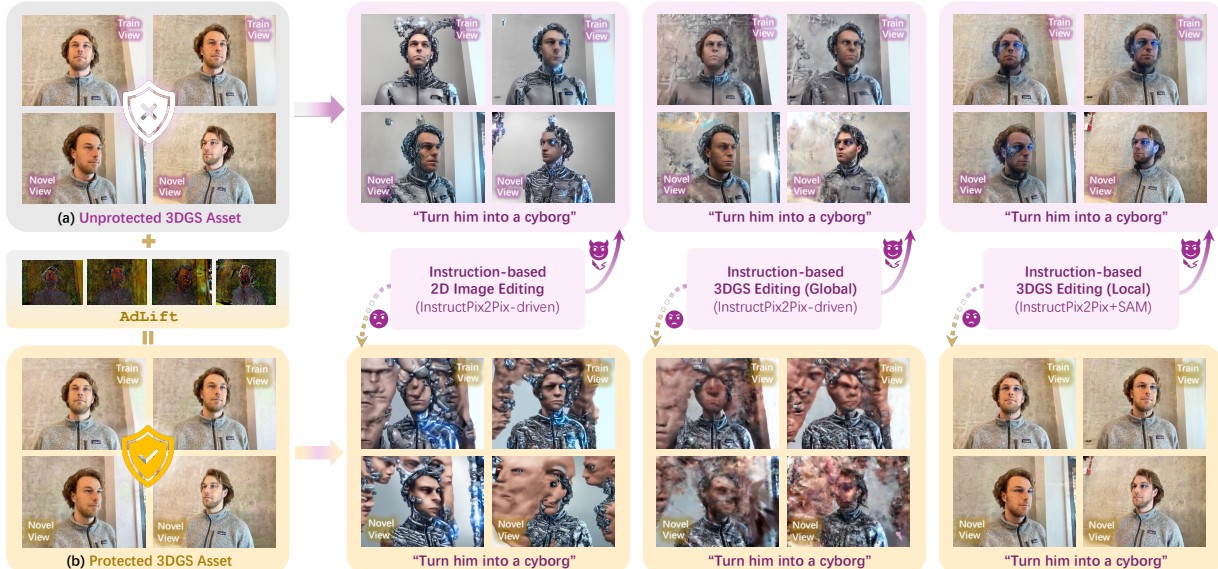

*Figure 1.* (a) Unprotected 3D Gaussian Splatting (3DGS) assets are exposed to serious risks of unauthorized modification and malicious tampering. (b) In this work, we introduce `AdLift`, a novel framework for proactively safeguarding 3DGS assets, effectively preventing unauthorized manipulation across arbitrary views and editing dimensions while preserving visual fidelity.

- ***View-generalizable protection*** *(i.e., suppressing editing effectiveness on the 3DGS itself as well as across arbitrary 2D rendering viewpoints).* A straightforward attempt is to fit a 3DGS model to multi-view 2D images that are individually protected with adversarial-perturbations-based 2D protection methods. However, such perturbations are inherently view-inconsistent. As illustrated in Figure 2, directly fitting may induce cross-view conflicts. Consequently, the resulting 3DGS suffers from underfitting and exhibits poor generalization to novel views.

- ***Balancing invisibility with protection capability*** *(i.e., preserving the appearance of the original 3DGS assets while retaining strong protection capability).* For 2D images, unediting methods typically learn adversarial perturbations on pixel grids (Goodfellow et al., 2014), where invisibility is enforced by strict budget. In contrast, 3DGS adopts fundamentally different representations (i.e., *Gaussian primitives with heterogeneous attributes* (Kerbl et al., 2023)), making it nontrivial to set a uniform budget that ensures both imperceptibility and effectiveness. Related passive protection works (e.g., Chen et al. (2025)) typically inject perturbations into Gaussian parameters via *soft invisibility regularization* over all or selected attributes. While such soft constraints maintain invisibility when embedding hidden messages, they become insufficient for learning imperceptible adversarial perturbations against powerful editing models. As illustrated in Figure 3, these approaches struggle to achieve a satisfactory trade-off between invisibility and adversarial attack strength.

In this work, we propose `AdLift` to lift adversarial perturbations for safeguarding 3DGS assets against instruction-driven editing. Instead of relying on soft constraints or imposing hard constraints directly on Gaussian parameters, `AdLift` enforces strictly bounded perturbations at the 2D rendering space and then lifts them into a group of "safeguard Gaussians" in 3D space, thus achieving both attack effectiveness and invisibility. To optimize the safeguard Gaussian parameters with hard constraint on rendering space, we introduce a tailored Lifted PGD (L-PGD), which includes two steps: (i) *gradient truncation* to enforce strict invisibility constraints, and (ii) *image-to-Gaussian fitting* to update the 3D parameters accordingly. Specifically, L-PGD truncates the gradient during back-propagation from the editing model to the Gaussians at the rendered image and applies projected gradients to strictly constrain the image-level perturbation. Then, the perturbation is backpropagated to the Gaussian parameters via an image-to-Gaussian fitting operation. L-PGD alternates between *gradient truncation* and *image-to-Gaussian fitting*, progressively seeking the optimal Gaussian-based adversarial perturbations with both consistent attack performance and invisibility across different views, thus yielding view-consistent adversarial-based editing safeguards that generalize effectively to novel views.

Empirically, we validate the protection performance of `AdLift` against instruction-driven 2D image editing and instruction-driven global/local 3DGS editing on multiple forward-facing and 360-degree scenes. Extensive experiments demonstrate the state-of-the-art protection performance of `AdLift`. *Typical examples are shown in Figure 1(b).* In summary, our main contributions are three-fold:

- We study the problem of active protection of 3DGS assets against instruction-driven editing. We identify two key challenges: *view-generalizable protection* and *balancing invisibility with protection capability.*

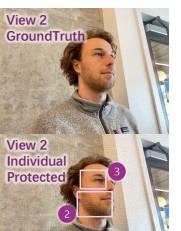 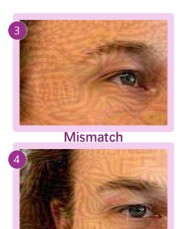 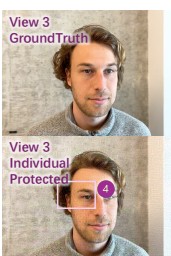 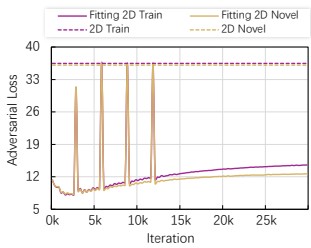

*Figure 2.* (*Left*): adversarial perturbations[1] learned separately for each view do not maintain multi-view consistency. (*Right*): multi-view inconsistencies cause underfitting, resulting in a large adversarial loss gap compared with 2D perturbations (denoted as 2D Train/Novel).

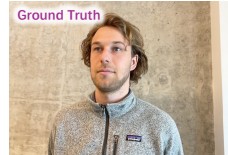 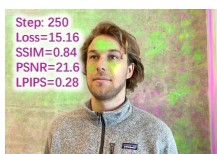 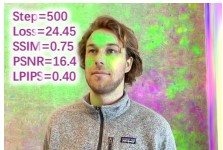 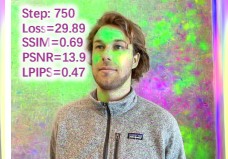 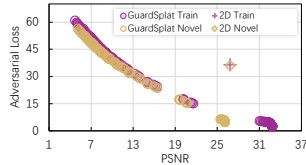

*Figure 3.* Perturbations[1] imposed on spherical-harmonic (SH) features (i.e., GuardSplat (Chen et al., 2025)) under soft invisibility constraints (e.g., SSIM) pose inherent challenges in balancing invisibility and attack strength. (*Left*): Perturbations learned by GuardSplat. (*Right*): PSNR-Adversarial Loss scatter plot over multiple runs under different attack-invisibility trade-off weights.

- We propose `AdLift`, which trains a group of dedicated safeguard Gaussians using a tailored Lifted PGD (L-PGD) that lifts strictly bounded perturbations from the 2D image space into the 3D Gaussian space to ensure both invisibility and protective effectiveness.

- Extensive experiments on multiple scenes and diverse instruction-driven editing tasks (2D/3D, global/local) demonstrate the general effectiveness of our `AdLift`.

## 2. Related Work

**Intellectual property (IP) protection for 3DGS.** Recent efforts have extended IP protection techniques to 3DGS, including watermarking (Huang et al., 2024; Chen et al., 2025; Li et al., 2025a; In et al., 2025; Jang et al., 2025; Huang et al., 2025) and steganography (Zhang et al., 2024; 2025; Ren et al., 2025). These approaches hide information within 3DGS assets by introducing imperceptible perturbations, serving as ownership watermarks or as steganographic payloads, that can be reliably extracted from both the 3D representation and its rendered 2D images. To preserve visual fidelity, their perturbations are typically restricted to specific Gaussian parameters. For example, GaussianMarker (Huang et al., 2024) densifies high-uncertainty regions by adding new Gaussians whose parameters are optimized to carry watermarks, while GuardSplat (Chen et al., 2025) perturbs only the spherical-harmonic (SH) features of existing Gaussians to embed watermarks without altering geometry. In both cases, watermark decoding objectives are jointly optimized with soft fidelity regularization (e.g., SSIM (Wang et al., 2004)) to mitigate artifacts and maintain rendering quality. However, existing watermarking- or steganography-based methods provide only *passive protection for 3DGS*

assets, e.g., they only enable ownership verification (Wang et al., 2021) by extracting watermark (or identity) messages from potentially leaked assets. Such passive schemes, however, are incapable of proactively defending against malicious editing enabled by instruction-driven 2D and 3DGS editing techniques (Brooks et al., 2023; Chen et al., 2024b).

**Edit Guard for 2D Image.** Recent studies have investigated editguard for 2D images (Chen et al., 2024c; Liang & Wu, 2023; Liu et al., 2024; Salman et al., 2023; Xue et al., 2024; Wang et al., 2024a; Teng et al., 2025). The core idea is to formulate unediting protection as an adversarial attack on editing models, injecting imperceptible perturbations into images so that downstream editing fails (e.g., produces extremely degraded output qualities). However, despite the success of editing guards for 2D images, their extension to 3DGS remains largely unexplored and introduces unique challenges, including the need to enforce *view-consistent perturbations for view-generalizable protection* and to *balance invisibility against protection effectiveness*.

## 3. Preliminary

**3D Gaussian Splatting (3DGS).** 3DGS (Kerbl et al., 2023) is an efficient explicit 3D representation that models a scene using anisotropic Gaussians, where each Gaussian $G$ is defined by mean $\mu \in \mathbb{R}^3$, covariance matrix $\Sigma$, color $c \in \mathbb{R}^3$, and opacity $\alpha \in \mathbb{R}$. A Gaussian centered at $\mu$ can be expressed as: $G(p) = \exp(-\frac{1}{2}(p-\mu)^\top \Sigma^{-1}(p-\mu))$, where $p$ denotes any 3D position. When rendering, the 3D Gaussians are projected into 2D and composited along each camera ray using a volume splatting method. Specifically, the color $C$ of a pixel is computed by blending $\mathcal{N}$ depth-ordered points: $C = \sum_{i \in \mathcal{N}} c_i \alpha_i \prod_{j=1}^{i-1}(1 - \alpha_j)$, where $c_i$ is the color estimated by the spherical harmonics (SH) coefficients of each Gaussian, and $\alpha_i$ is given by evaluating a 2D Gaussian with

---

[1]We here conduct an untargeted attack in Equation (7); higher loss means a stronger attack strength.

covariance $\Sigma'$ (Yifan et al., 2019) multiplied with a per-point opacity. By integrating all Gaussians $G$, the 3DGS model $\mathcal{G}$ forms a compact yet expressive representation of 3D scenes.

**Instruction-driven Image Editing.** Latent Diffusion Models (LDMs) (Rombach et al., 2022) serve as the backbone for modern image editing. In LDM, an Variational Autoencoder (VAE), consisting of an encoder $\mathcal{E}$ and a decoder $\mathcal{D}$, first maps an image $x$ into a latent code $z = \mathcal{E}(x)$. In the latent space, a diffusion model $\epsilon_\theta$ then learns to denoise the noised latent $z_t$ at each time step $t$, conditioned on an embedding $c_\theta(y)$ derived from the input $y$. The standard LDM training objective is: $\mathcal{L}_{\text{LDM}} = \mathbb{E}_{z,y,\epsilon,t}[\|\epsilon - \epsilon_\theta(z_t, t, c_\theta(y))\|_2^2]$, where $z_t$ is the latent corrupted to step $t$, $\epsilon$ is the Gaussian noise sample. Furthermore, **InstructPix2Pix (IP2P)** and following works (Brooks et al., 2023; Zhang et al., 2023; Podell et al., 2023) extends LDMs to *directly follow human editing instructions*. IP2P fine-tunes the LDM on triplets $(x^{\text{src}}, y, x^{\text{edit}})$, where $x^{\text{src}}$ is the source image, $y$ is the instruction, and $x^{\text{edit}}$ is the corresponding edited image, and the denoiser $\epsilon_\theta$ is conditioned jointly on the original latent $z_0 = \mathcal{E}(x^{\text{src}})$ and the instruction embedding $c_\theta(y)$. Through this fine-tuning, *IP2P enables users to perform direct, instruction-driven edits on input images, yielding results that faithfully follow human-provided instructions.*

**Instruction-driven 3DGS editing.** Given a source 3DGS model $\mathcal{G}^{\text{src}}$ and an editing instruction $y$, the objective of instruction-guided 3DGS editing is to transform $\mathcal{G}^{\text{src}}$ into an edited version $\mathcal{G}^{\text{edit}}$ that complies with $y$. Current approaches achieve this by leveraging 2D editing techniques (Lee et al., 2025; Chen et al., 2024b; Wang et al., 2024b; Vachha & Haque, 2024). Specifically, $\mathcal{G}^{\text{src}}$ is first rendered from multiple training views $\mathcal{V}^t = \{v_i^t\}_{i=1}^{N_t}$ to produce a set of source images $\mathcal{I}^{\text{src}} = \{x_i^t = \mathcal{R}(\mathcal{G}, v_i^t)\}_{i=1}^{N_t}$. A 2D editing model (e.g., IP2P) then modifies $\mathcal{I}^{\text{src}}$ into edited images $\mathcal{I}^{\text{edit}}$ according to the specified instruction $y$. These edited images provide supervision for updating $\mathcal{G}^{\text{src}}$, yielding the edited $\mathcal{G}^{\text{edit}}$ by minimizing the editing loss $\mathcal{L}_{\text{Edit}}$ across all training views $\mathcal{V}$: $\mathcal{G}^{\text{edit}} = \arg\min_{\mathcal{G}} \sum_{v^t \in \mathcal{V}} \mathcal{L}_{\text{Edit}}(\mathcal{R}(\mathcal{G}, v^t), \mathcal{I}^{\text{edit}})$, where $\mathcal{R}$ denotes the rendering function that projects a 3DGS into an image under view $v$. Besides, multi-view consistency constraints (Chen et al., 2024b) can be additionally applied for preventing low-quality results caused by inconsistencies in the 2D guidance images.

# 4. Lifting Adversarial Perturbations on 3DGS

## 4.1. Problem Statement

**Assumptions on defender's capability.** We assume the defender has *white-box access* to a surrogate instruction-driven editing model used to simulate attacks during protection design, but has *no control over the actual editing models and editing instructions adopted by potential attackers*. This allows us to construct the protection mechanism via white-

box adversarial optimization, and subsequently test its transferability against unseen editing models and instructions. Furthermore, we assume the defender has access to the original 3DGS asset $\mathcal{G}^{\text{raw}}$ trained on multi-view ground-truth images $\mathcal{I}^t = \{x_v^t | v \in \mathcal{V}^t\}$, where $\mathcal{V}^t = \{v_i^t\}_{i=1}^{N_t}$.

**Protection objective.** Our objective is to design a protection mechanism for transform $\mathcal{G}^{\text{raw}}$ into $\mathcal{G}^{\text{prot}}$ through slight modifications that **preserve visual fidelity** while enforcing the following properties:

- **2D protection:** Any rendered view from $\mathcal{G}^{\text{prot}}$ remains robust against instruction-driven 2D image editing methods, thereby preventing view-wise manipulations.
- **3D protection:** The entire asset $\mathcal{G}^{\text{prot}}$ cannot be effectively manipulated by instruction-driven 3DGS editing, ensuring the integrity of the original 3DGS asset.

## 4.2. Rethinking Gaussians-based Perturbations

This task can be regarded as a natural 3D extension of *edit guards* in the 2D image domain, which typically construct *adversarial perturbations* (Goodfellow et al., 2014; Madry et al., 2017; Liang et al., 2023) to prevent image editing by instruction-tuned diffusion models (Brooks et al., 2023). Specifically, for a single 2D image $x^t$, the protected image $x^{\text{prot}}$ can be derived via optimize the following objective:

$$x^{\text{prot}} = \arg\min_x \ \mathcal{L}_{\text{adv}}(x, y) \quad \text{s.t.} \ \|x - x^t\|_\infty \leq \eta, \quad (1)$$

where $\mathcal{L}_{\text{adv}}(\cdot, y)$ is the adversarial loss[2] enforcing resistance against the editing model under instruction $y$, $\|\cdot\|_\infty$ represents the $\infty$-norm which measures perturbation magnitude, and $\eta$ controls the perturbation budget to ensure imperceptibility. Intuitively, the objective seeks a perturbed $x^{\text{prot}}$ that remains visually indistinguishable from $x$ while disrupting the effect of instruction-driven editing.

However, directly extending 2D image protection to 3DGS leaves a critical gap. A straightforward attempt is to fitting the protected model $\mathcal{G}^{\text{prot}}$ on multiple 2D protected training views $\{x_{v^t}^{\text{prot}}\}, v^t \in \mathcal{V}^t$:

$$\mathcal{G}^{\text{prot}} = \arg\min_{\mathcal{G}} \ \sum_{v^t \in \mathcal{V}^t} \mathcal{L}_{\text{rec}}(\mathcal{R}(\mathcal{G}, v^t), x_{v^t}^{\text{prot}}), \quad (2)$$

where $\mathcal{R}(\mathcal{G}, v^t)$ is its rendering under view $v^t$, and $\mathcal{L}_{\text{rec}}$ is a photometric reconstruction loss (like SSIM (Wang et al., 2004)). Unfortunately, independently optimized perturbations on are not view-consistent, leading to *cross-view conflicts*, as illustrated in Figure 2. As a result, training $\mathcal{G}^{\text{prot}}$ causes underfitting even on training views and severely degrades generalization to unseen viewpoints.

Considering the natively view-continuity of Gaussians, several works (Huang et al., 2024; Chen et al., 2025) use a group of Gaussians to represent perturbations and train them from the 3D space or dynamically from 2D views, rather

---

[2]We assume smaller $\mathcal{L}_{\text{adv}}$ indicates better attack performance.

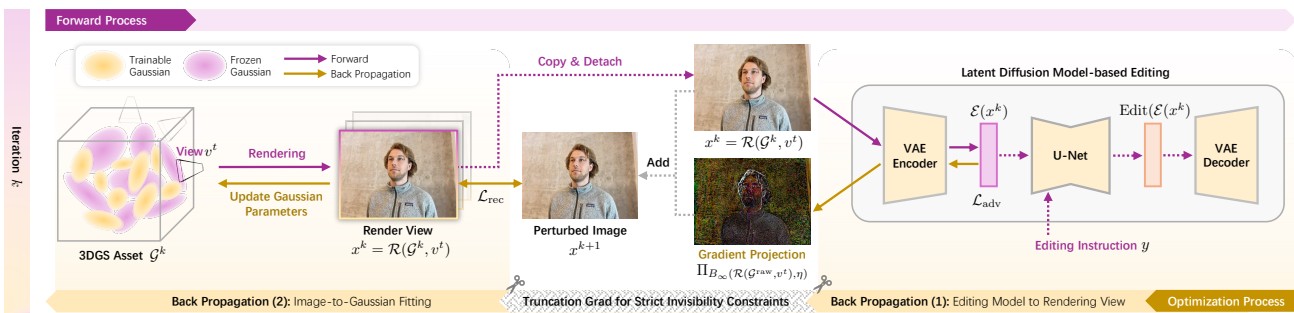

*Figure 4.* The proposed method `AdLift`, where a tailored Lifted PGD (L-PGD) is used for learning Gaussians-represented adversarial perturbations against Instruction-based editing models. L-PGD alternates between *gradient truncation*, which enforces invisibility in the image domain, and *image-to-Gaussian fitting*, which propagates the constrained perturbations back into the 3D space.

than directly fitting in-consistent 2D supervisions. However, unlike the directly fitting where objectives $\{x_{v^t}^{\text{prot}}\}$ already satisfy both the imperceptible and protection requirements, *a critical challenge for training Gaussian-based perturbation is how to preserve the appearance consistency between $\mathcal{G}^{prot}$ and $\mathcal{G}^{raw}$ while retaining strong protection capability.*

Gaussians-based perturbations are not easy to set a suitable perturbation bound. In Equation (1), unediting methods for 2D images typically learn adversarial perturbations on pixel grids, with invisibility enforced by predefined budget $\eta$ constraints and projection operations (Madry et al., 2017). In contrast, 3DGS employs Gaussian primitives with a fundamentally different representation, making it nontrivial to design invisible yet effective perturbations[3]. Existing works which try to learn Gaussian-based perturbations (Huang et al., 2024; Chen et al., 2025) typically add **soft constraints** via invisible regularization on all or selected Gaussian attributes for maintaining perturbation imperceptibility. Although being effective for hidden messages, these approaches fail to strike a satisfactory balance between imperceptibility and adversarial attack strength for instruction-based editing models, as illustrated in Figure 3.

### 4.3. Rendering-space Strictly Bound for Gaussians-based Perturbations

In order to balance imperceptibility and protection strength of 3DGS perturbations, we propose to impose **strictly bounded constraints on rendering view space**. Formally, the objective of learning 3DGS adversarial perturbations can be formulated as:

$$\mathcal{G}^{\text{prot}} = \arg\min_{\mathcal{G}} \sum_{v^t \in \mathcal{V}^t} \mathcal{L}_{\text{adv}}(\mathcal{R}(\mathcal{G}, v^t), y)$$
$$\text{s.t. } \|\mathcal{R}(\mathcal{G}, v^t) - \mathcal{R}(\mathcal{G}^{\text{raw}}, v^t)\|_\infty \leq \eta, \ \forall v^t \in \mathcal{V}^t, \quad (3)$$

where we set a strict upper bound $\eta$ (i.e., the perturbation budget) via the $\infty$-norm on all the training views. Intuitively, by enforcing perturbation bounds in the image domain, our approach circumvents the instability of soft regularizations

---

[3]Heterogeneous attributes (e.g., position, scale, color, opacity) make uniform budgets ineffective.

and the difficulty of hard Gaussian-parameter constraints, thus ensuring perceptually meaningful control and consistent invisibility across views.

### 4.4. Lifted Projected Gradient Descent

Solving the optimization problem in Equation (3) is not straightforward. Considering the $k$-th iteration and for a given training view $v^t$, the update rule of gradient descent-based solver for Equation (3) can be formulated as:

$$\mathcal{G}^{k+1} = \Pi_{\mathcal{C}}(\mathcal{G}^k - \alpha \cdot \nabla_{\mathcal{G}} \mathcal{L}_{\text{adv}}(\mathcal{R}(\mathcal{G}^k, v^t), y)), \quad (4)$$

where $\Pi_{\mathcal{C}}$ denotes projection onto the feasible set $\mathcal{C} = \{\mathcal{G} : \|\mathcal{R}(\mathcal{G}, v^t) - \mathcal{R}(\mathcal{G}^{\text{raw}}, v^t)\|_\infty \leq \eta, \ \forall v^t \in \mathcal{V}^t\}$. Equation (4) requires *(i) updating Gaussian parameters in the 3D space to minimize the adversarial loss*, while *(ii) simultaneously enforcing hard constraints in the image domain across multiple rendered views to ensure visual quality*. The discrepancy between the optimization variables (3D Gaussian primitives) and the constraint space (multi-view 2D projections) complicates the problem.

To solve Equation (3), we introduce a tailored Lifted PGD (L-PGD). As shown in Figure 4, L-PGD consists of two main steps: *(i) gradient truncation to enforce strict invisibility constraints*, and *(ii) image-to-Gaussian fitting to update the 3D parameters accordingly.*

**Gradient truncation for strict invisibility constraints.** Although feasible set $\mathcal{C}$ is *implicitly* defined in the 3D Gaussian parameter space, its explicit characterization becomes available in the rendered image space. This means that while we may not know the precise form of $\mathcal{C}$ in Gaussian-parameter space, we can still approximate it by evaluating surrogate renderings from the prospective update of $\mathcal{G}$ that achieves a smaller adversarial loss. Let $x^k = \mathcal{R}(\mathcal{G}^k, v^t)$ denote the current rendered image, we truncate the gradient in Equation (4) on the image domain, so the projected update becomes:

$$x^{k+1} = \Pi_{B_\infty(x_{v_t}^{\text{ref}}, \eta)} \left[ x^k - \alpha \cdot \text{sign}(\nabla_x \mathcal{L}_{\text{adv}}(x^k, y)) \right], \quad (5)$$

where $B_\infty(x_{v_t}^{\text{ref}}, \eta)$ denotes the $\ell_\infty$ ball of radius $\eta$ centered at $x_{v_t}^{\text{ref}} = \mathcal{R}(\mathcal{G}^{\text{raw}}, v^t)$. This guarantees that the updated rendering $x^{k+1}$ strictly satisfies the hard imperceptibility constraint, i.e., it can be regarded as the rendered

image of a hypothetical $(k + 1)$-th 3DGS model $\hat{\mathcal{G}}^{k+1}$, with $\mathcal{R}(\hat{\mathcal{G}}^{k+1}, v^t) = x^{k+1}$, where $\hat{\mathcal{G}}^{k+1}$ necessarily lies within the feasible set $\mathcal{C}$ and has smaller adversarial loss.

**Image-to-Gaussian fitting.** Given the current 3DGS model $\mathcal{G}^k$ and the surrogate rendered image $x^{k+1} = \mathcal{R}(\hat{\mathcal{G}}^{k+1}, v^t)$ from a hypothetical next-step Gaussian model $\hat{\mathcal{G}}^{k+1}$ that achieves a smaller adversarial loss, we treat $x^{k+1}$ as a supervision signal. The Gaussian parameters of $\mathcal{G}^k$ are then updated by fitting its rendering $\mathcal{R}(\mathcal{G}^k, v^t)$ to $x^{k+1}$. Formally, given the training view $v^t$, the update rule of the image-to-Gaussian fitting step can be formulated as:

$$\mathcal{G}^{k+1} = \mathcal{G}^k - \beta \cdot \nabla_{\mathcal{G}} \mathcal{L}_{\text{rec}}(\mathcal{R}(\mathcal{G}^k, v^t), x^{k+1}), \quad (6)$$

where $\beta$ is the learning rate for the image-to-Gaussian fitting step and $\mathcal{L}_{\text{rec}}$ is a reconstruction loss. In this way, $\mathcal{G}^{k+1}$ is optimized toward the hypothetical next-step Gaussians, thereby remaining within the feasible set while achieving a smaller adversarial loss.

**Alternating optimization.** The safeguard Gaussians $\mathcal{G}^{\text{prot}}$ are jointly and progressively optimized across all training views $v^t \in \mathcal{V}^t$. L-PGD alternates between *gradient truncation*, which enforces invisibility in the image domain, and *image-to-Gaussian fitting*, which propagates the constrained perturbations back into the 3D space. Through this alternating process, $\mathcal{G}^{\text{prot}}$ is optimized to minimize the adversarial loss while simultaneously satisfying strict imperceptibility bounds across multiple viewpoints. The overall algorithm for training AdLift via L-PGD is shown in Algorithm 1.

*Intuitively, L-PGD optimizes safeguard Gaussians $\mathcal{G}^{\text{prot}}$ by lifting strictly bounded and multi-view consistent adversarial perturbations from the 2D rendered image domain into the intrinsic 3D Gaussian space.* The dynamic alternating optimization procedure in L-PGD avoids the need to manually assign heterogeneous budgets on Gaussian parameters, which are difficult to calibrate and may cause visible distortions if mis-specified. Instead, the *rendering-space bound* ensures that every intermediate update is constrained to remain visually indistinguishable from the original views, while the *image-to-Gaussian fitting* propagates these perturbations back into 3D Gaussian parameters. In this way, the safeguard Gaussians inherit the adversarial protection strength observed in individual 2D cases, but extend it coherently across all viewpoints, thereby ensuring continuous and cross-view consistent protection.

### 4.5. Adversarial Objectives for Resisting Editing Model

L-PGD is compatible with various adversarial loss functions $\mathcal{L}_{\text{adv}}$, enabling the design of different AdLift variants tailored to specific requirements. We illustrate several representative cases of adversarial attacks on instruction-based 2D and 3D global/local editing tasks. As directly attacking the diffusion model itself has proven challenging (Chen et al., 2024a; Xue & Chen, 2024), we instead target other

---

**Algorithm 1** Training AdLift via Lifted PGD (L-PGD).

1: **Input:** Training views $\mathcal{V}^t$; unprotected 3DGS asset $\mathcal{G}^{\text{raw}}$ and trainable asset with safeguard $\mathcal{G}$; Total iterations $E$; Gradient truncation iterations $K_p$; Image-to-Gaussian fitting iterations $K_l$; Learning rates $\alpha, \beta$; Perturbation budget $\eta$.
2: **for** $k = 1$ to $E - 1$ **do**
3:     Sample a training view $v^t \sim \mathcal{V}^t$;
4:     Render: $x^{k(1)} = \mathcal{R}(\mathcal{G}^k, v^t)$, $x_{v_t}^{\text{ref}} = \mathcal{R}(\mathcal{G}^{\text{raw}}, v^t)$;
5:     **for** $i = 1$ to $K_p - 1$ **do**
6:         $x^{k(i+1)} = \Pi_{B_\infty(x_{v_t}^{\text{ref}}, \eta)}[x^{k(i)} - \alpha \cdot \text{sign}(\nabla_x \mathcal{L}_{\text{adv}}(x^{k(i)}))]$;
7:     **end for**
8:     Let $x^{k+1} = x^{k(K_p)}$, and $\mathcal{G}^{k(1)} = \mathcal{G}^k$;
9:     **for** $j = 1$ to $K_l - 1$ **do**
10:         $\mathcal{G}^{k(j+1)} = \mathcal{G}^{k(j)} - \beta \cdot \nabla_{\mathcal{G}} \mathcal{L}_{\text{rec}}(\mathcal{R}(\mathcal{G}^{k(j)}, v^t), x^{k+1})$;
11:     **end for**
12:     Let $\mathcal{G}^{k+1} = \mathcal{G}^{k(K_l)}$;
13: **end for**
14: **Output:** Trained safeguard Gaussians $\mathcal{G}^{\text{prot}} = \mathcal{G}^E$.

---

*instruction-agnostic* components integrated in instruction-driven editing pipelines:

- *Attack VAE (Untargeted)*: Encourage latent code of the rendered image $\mathcal{R}(\mathcal{G}, v^t)$ to deviate from that of the raw rendering. This leads to artifacts that hinder faithful instruction-driven editing:

$$\mathcal{L}_{\text{VU}} = -\|\mathcal{E}(\mathcal{R}(\mathcal{G}, v^t)) - \mathcal{E}(\mathcal{R}(\mathcal{G}^{\text{raw}}, v^t))\|_2^2. \quad (7)$$

- *Attack VAE (Targeted)*: Also referred to as *Textural Loss* (Salman et al., 2023; Liang & Wu, 2023; Shan et al., 2023; Xue et al., 2024). This loss aligns the latent code of $\mathcal{R}(\mathcal{G}, v^t)$ with that of a chosen target image or pattern $x^{\text{target}}$, injecting irrelevant textures that compromise realism and misguide the editing process:

$$\mathcal{L}_{\text{VT}} = \|\mathcal{E}(\mathcal{R}(\mathcal{G}, v^t)) - \mathcal{E}(x^{\text{target}})\|_2^2. \quad (8)$$

- *Attack Segment-Anything (SAM)*: Given a rendered image $\mathcal{R}(\mathcal{G}, v^t)$, SAM (Kirillov et al., 2023) $\mathcal{S}(\mathcal{R}(\mathcal{G}, v^t), b)$ predicts a mask conditioned on bounding box $b$, which is widely adopted in 3DGS editing for local region localization. Thus, we enforce the predicted mask to match a target mask $m^{\text{target}}$ for misguiding segmentation, resulting in unnatural boundaries and corrupted local edits:

$$\mathcal{L}_{\text{ST}} = \mathcal{L}_{\text{BCE}}(\mathcal{S}(\mathcal{R}(\mathcal{G}, v^t), b), m^{\text{target}}) \\ + \lambda_{\text{Dice}} \mathcal{L}_{\text{Dice}}(\mathcal{S}(\mathcal{R}(\mathcal{G}, v^t), b), m^{\text{target}}), \quad (9)$$

where $m^{\text{target}}$ is the desired mask and $\lambda_{\text{Dice}}$ controls the relative weight of the two terms.

## 5. Experiments

**Datasets:** We follow 3DGS editing works (Chen et al., 2024b) to conduct experiments on two forward-facing scenes from Haque et al. (2023) and (Wang et al., 2023) and two 360degree scenes from Haque et al. (2023) and Yao et al. (2020). **Edit baselines:** 2D Editing: Instruction-Pix2Pix (IP2P) (Brooks et al., 2023), SDEdit (Meng et al.,

*Table 1.* Quantitative results of *instruction-based image editing* (averaged across four scenes). We report results on both *training* and *novel views*. The best/second best results are denoted as **bold**/underline. More results are shown in Appendix C.

| | Method | Invisibility | | | Protection Quality | |
|---|---|---|---|---|---|---|
| | | SSIM(↑) | PSNR(↑) | LPIPS(↓) | CLIP$_d$(↓) | CLIP$_s$(↓) |
| Training views | No Protection | 0.9215 | 30.4812 | 0.0805 | 0.1875 | 0.8892 |
| | Fit2D-VU | 0.7557 | 27.0716 | 0.2566 | 0.1797 | 0.8368 |
| | Fit2D-VT | **0.8232** | **27.9244** | **0.1433** | 0.1752 | 0.8566 |
| | AdLift-VU | 0.7751 | 27.7292 | 0.2528 | 0.1787 | 0.8139 |
| | AdLift-VT | 0.8133 | 27.8060 | 0.1670 | 0.1748 | 0.8370 |
| | AdLift*-VU | 0.7280 | 27.4019 | 0.3069 | 0.1806 | **0.7889** |
| | AdLift*-VT | 0.7896 | 27.6083 | 0.1848 | **0.1731** | 0.8258 |
| Novel views | No Protection | 0.8061 | 25.0546 | 0.1322 | 0.1901 | 0.8819 |
| | Fit2D-VU | 0.6478 | 23.7012 | 0.2882 | 0.1825 | 0.8488 |
| | Fit2D-VT | 0.7168 | **24.3197** | 0.1869 | 0.1798 | 0.8592 |
| | AdLift-VU | 0.6802 | 23.7918 | 0.2841 | 0.1799 | 0.8181 |
| | AdLift-VT | **0.7182** | 23.8981 | 0.2095 | 0.1782 | 0.8366 |
| | AdLift*-VU | 0.6374 | 23.9092 | 0.3330 | 0.1802 | **0.8087** |
| | AdLift*-VT | 0.7008 | 24.0815 | 0.2240 | **0.1758** | 0.8296 |

2021), MagicBrush (Zhang et al., 2023) and SDXL-IP2P (Podell et al., 2023). 3D Editing: Instruct-GS2GS (Vachha & Haque, 2024; Haque et al., 2023) and DGE (Chen et al., 2024b). **Protect baselines:** We compare (i): asset without protection, (ii) soft constraints: GuardSplat (Chen et al., 2025) and GaussianMarker (Huang et al., 2024), and (iii) fitting the 3DGS on 2D protected images (i.e., Equation (2), denoted as Fit2D). **Evaluation metrics:** *Invisibility:* We evaluate the visual similarity of views rendered from the models and ground truth using SSIM (Wang et al., 2004), PSNR, and LPIPS (Zhang et al., 2018). *Protection Capability:* We use two metrics for measuring protection effectiveness: CLIP Text-Image Directional Similarity (CLIP$_d$) and CLIP Image Similarity (CLIP$_s$). **Implementation for AdLift:** We train AdLift by using IP2P as the surrogate editing model. We have two ways for initialization: (i) copy all of the original gaussians for the unprotected asset $\mathcal{G}^{raw}$, and use these new gaussians as trainable parameters while fix all of the original Gaussians. We denote this as AdLift. (ii) we can also continue training our AdLift on the 3DGS which directly fitting on independent 2D adversarial perturbations. We denote it as AdLift*. Besides, we mark different adversarial objectives after the method name[4]. **Implementation for editing:** We use default hyperparameters for each editing method. Editing instructions are parts collected from previous works (Wu et al., 2024) and further extended by using GPT-5 and manually reviewed. All experiments are conducted on NVIDIA RTX 4090 GPU. More details are shown in Appendix G.

## 5.1. Evaluation on InstructionPix2Pix

We show *visualization of protection perturbations* in Figure 5. The adversarial noise learned by AdLift remains spatially coherent across different viewpoints. Semantic regions such as the face and hair are perturbed in a consistent manner rather than producing mismatched or fragmented

---

[4] For example, AdLift-VU means training AdLift by using the untargeted VAE loss in Equation (7).

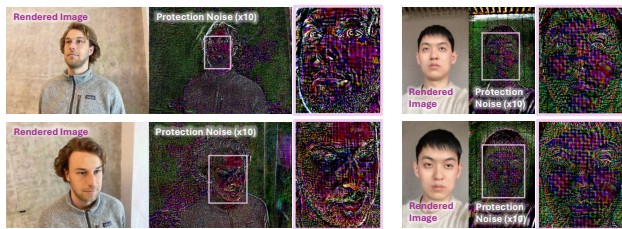

*Figure 5.* Protection perturbations learned by AdLift*-VU.

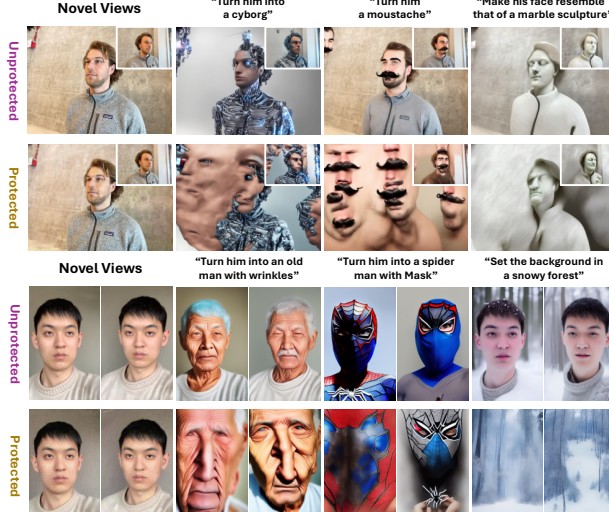

*Figure 6.* Qualitative results of *instruction-based image editing* on two face scenes. We show the comparison of editing results of unprotected 3D assets and 3D assets protected by AdLift*-VU.

patterns. *Qualitative results* for protection are shown in Figure 6. We present comparisons of instruction-driven editing applied to both unprotected and AdLift-protected 3DGS assets. Compared to unprotected, assets protected by AdLift effectively resist editing either on training views and novel views, producing unrealistic editing results by following the instructions. Importantly, this protection generalizes from training to novel views, demonstrating that our lifted perturbations enforce view-consistent and generalizable safeguards. Besides, we present *quantitative results* in Table 1. From the results, our AdLift consistently achieves stronger protection (lower CLIP$_d$ and CLIP$_s$) while maintaining competitive invisibility metrics (SSIM, PSNR, LPIPS) across both training and novel views.

## 5.2. Transferability to Unseen 2D and 3DGS Editing

We additionally evaluate whether AdLift, which aims at lifting 2D adversarial perturbations into the 3D Gaussian space, can inherit the transferability in 2D adversarial protection (Chen et al., 2024c). We test AdLift against three unseen 2D editing methods: SDEdit (Meng et al., 2021) (with different editing pipelines) MagicBrush (Zhang et al., 2023) (fine-tuned variants) and SDXL-IP2P (Podell et al., 2023) (different base model), and two 3DGS editing methods: Instruct-GS2GS (Vachha & Haque, 2024; Haque et al., 2023) and DGE (Chen et al., 2024b). *Quantitative results*

*Table 2.* Transferability to unseen 2D and 3DGS editing (novel view). The best results are denoted as **bold**. More results in Appendix F.

| | Method | Surrogate | | Unseen 2D Editing | | | | | | | | Unseen 3D Editing | | | |
| | | IP2P | | MagicBrush | | SDEdit | | SDXL-IP2P | | Instruct-GS2GS | | DGE | |
| | | CLIP$_d$($\downarrow$) | CLIP$_s$($\downarrow$) | CLIP$_d$($\downarrow$) | CLIP$_s$($\downarrow$) | CLIP$_d$($\downarrow$) | CLIP$_s$($\downarrow$) | CLIP$_d$($\downarrow$) | CLIP$_s$($\downarrow$) | CLIP$_d$($\downarrow$) | CLIP$_s$($\downarrow$) | CLIP$_d$($\downarrow$) | CLIP$_s$($\downarrow$) |
|---|---|---|---|---|---|---|---|---|---|---|---|---|---|
| Face | No Protection | 0.1704 | 0.8730 | 0.1769 | 0.8635 | 0.1708 | 0.7727 | 0.1140 | 0.7012 | 0.1108 | — | 0.1218 | — |
| | AdLift*-VT | 0.1595 | 0.8285 | 0.1607 | 0.8148 | 0.1751 | **0.7353** | **0.1081** | 0.6693 | 0.1031 | 0.8682 | **0.1166** | 0.9298 |
| | AdLift*-VU | **0.1586** | **0.7786** | **0.1475** | **0.7659** | **0.1554** | 0.7397 | 0.1094 | **0.6481** | **0.0857** | 0.8046 | 0.1200 | **0.9029** |
| Fangzhou | No Protection | 0.1938 | 0.9053 | 0.1812 | 0.8566 | 0.1220 | 0.7578 | 0.0738 | 0.7496 | 0.1324 | — | 0.1713 | — |
| | AdLift*-VT | **0.1788** | 0.8148 | **0.1581** | 0.7998 | **0.1194** | **0.6755** | 0.0715 | 0.7331 | **0.1119** | 0.8731 | **0.1432** | 0.8875 |
| | AdLift*-VU | 0.1791 | **0.8037** | 0.1605 | **0.7608** | 0.1292 | 0.7300 | **0.0695** | 0.7321 | 0.1160 | **0.8259** | 0.1522 | **0.8630** |

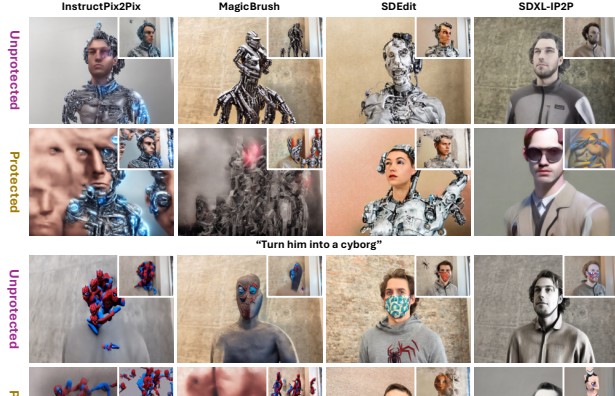

*Figure 7.* Transferability to unseen 2D editing.

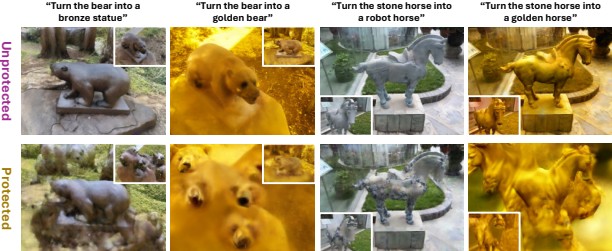

*Figure 8.* Transferability to unseen 3DGS editing.

*Table 3.* Comparison at similar PSNR-level (Face). Best results are highlighted in **bold**. More results are shown in Appendix B.

| Method | Training Views | | | Novel Views | | |
| | PSNR($\uparrow$) | CLIP$_d$($\downarrow$) | CLIP$_s$($\downarrow$) | PSNR($\uparrow$) | CLIP$_d$($\downarrow$) | CLIP$_s$($\downarrow$) |
|---|---|---|---|---|---|---|
| No Protection | 32.6436 | 0.1685 | 0.8852 | 26.0390 | 0.1704 | 0.8730 |
| GuardSplat | 31.7765 | 0.1649 | 0.8769 | 25.7744 | 0.1654 | 0.8666 |
| GuardSplat-(PC) | 25.2876 | 0.1605 | 0.8592 | 23.8763 | 0.1667 | 0.8575 |
| GuardSplat-(Full) | 25.1306 | 0.1617 | 0.8581 | 23.7877 | 0.1658 | 0.8535 |
| GaussianMarker | 28.8343 | 0.1596 | 0.8590 | 24.4911 | 0.1628 | 0.8578 |
| AdLift*-VU | 28.3963 | **0.1554** | **0.7620** | 25.6115 | **0.1586** | **0.7786** |

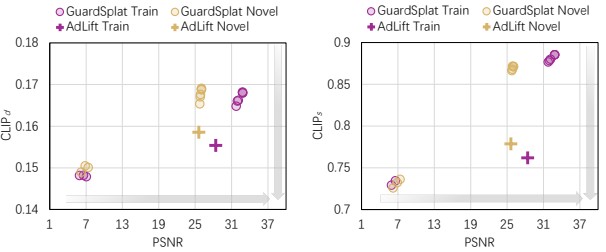

*Figure 9.* Results of CLIP$_d$($\downarrow$)/CLIP$_s$($\downarrow$) – PSNR($\uparrow$) on AdLift-VU and GuardSplat with different training hyperparameters.

are reported in Table 2 and *qualitative results* are shown in Figures 7 and 8. Empirically, across all unseen editing models, AdLift degrades editing quality compared to the unprotected version, demonstrating that the protective effect generalizes beyond the surrogate model.

### 5.3. Comparison with Soft Constraint Baselines

**Trade-off between invisibility and protection.** We compare our AdLift with GuardSplat (Chen et al., 2025), which perturbs the spherical-harmonic (SH) features of Gaussians under soft invisibility regularization. As shown in Figure 9, GuardSplat struggles to balance imperceptibility and protection: achieving stronger protection inevitably sacrifices imperceptibility, while preserving fidelity leads to weak protection. In contrast, our AdLift enforces strict bounds in the rendering space, reaching a much better trade-off (lower CLIP and higher PSNR simultaneously).

**Comparison at the matched visual quality.** We adapt GuardSplat and GaussianMarker (Huang et al., 2024) and compare them against AdLift under matched visual qual-

ity (similar PSNR). We preserve each method's original fidelity-preserving loss but replace their watermark decoding loss with our adversarial objective ($\mathcal{L}_{VU}$) so that all methods are optimized toward the same protection goal. We also consider two variants of GuardSplat: GuardSplat-(PC) (perturbing Positions and Covariance features) and GuardSplat-(Full) (perturbing all features). From the results in Table 3, AdLift consistently achieves stronger protection performance under similar visual quality.

### 5.4. More Results and Analysis

Due to limited space, more results are shown in Appendices, including but not limited to: robustness to purification (Appendix F), hyperparameter analysis (Appendices D and E).

## 6. Conclusion

In this work, we explore safeguarding 3DGS assets against instruction-driven editing. For achieving view-generalizable protection and maintaining a delicate balance between invisibility and protection strength, we introduce AdLift, a novel framework that trains safeguard Gaussians via a tailored Lifted PGD strategy, lifting bounded perturbations from the 2D image domain into the 3D Gaussian space. Extensive experiments on multiple 3DGS scenes and a wide range of editing tasks demonstrate the effectiveness of the proposed AdLift, marking a significant step toward securing 3DGS assets against unauthorized editing.

## Acknowledgements

The authors would like to thank the anonymous reviewers for their insightful and constructive comments. Ziming Hong is supported by JD Technology Scholarship for Postgraduate Research in Artificial Intelligence No. SC4103. Mingming Gong is supported by ARC DP240102088 and WIS-MBZUAI 142571. Bo Han is supported by RGC Young Collaborative Research Grant No. C2005-24Y and RGC General Research Fund No. 12200725. Tongliang Liu is partially supported by the following Australian Research Council projects: FT220100318, DP260102466, DP220102121, LP220100527, LP220200949.

## Impact Statement

This work addresses the rising risks of unauthorized editing and malicious manipulation of 3DGS assets. We propose an active protection framework to safeguard the intellectual property of 3D content creators and prevent the misuse of generative editing models. Although our method leverages adversarial perturbations, they are employed solely for defensive purposes rather than offensive attacks. We disclose our findings to ensure transparency and stress that the method should only be applied in ethically and legally appropriate scenarios for protecting digital assets.

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

## Appendices:

## A. Training Dynamics

Figure 10 compares the training dynamics of directly fitting 2D views individually protected with adversarial perturbations (denoted as Fit2D) and our AdLift variants, where higher adversarial loss indicates stronger attack strength. As discussed in the introduction, directly optimizing perturbations in the 2D domain suffers from view-specific conflicts, which cause underfitting on training views and poor generalization to novel views. As a result, Fit2D yields only limited improvement in adversarial loss even after prolonged training. By contrast, our Lifted-PGD-based strategy quickly enhances the adversarial loss on both training and novel views, demonstrating its ability to achieve view-generalizable protection. Moreover, initializing AdLift* from a pretrained Fit2D model yields consistently higher protection strength compared to applying AdLift directly on raw assets, highlighting the benefit of warm-starting with 2D guidance. Nevertheless, even without such initialization, AdLift significantly outperforms direct 2D fitting, confirming that lifting adversarial perturbations into the 3D Gaussian space is crucial for achieving both effectiveness and cross-view consistency.

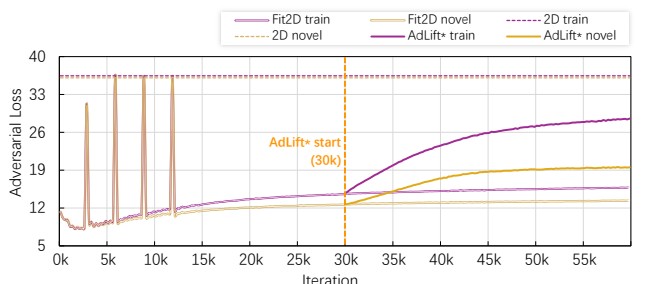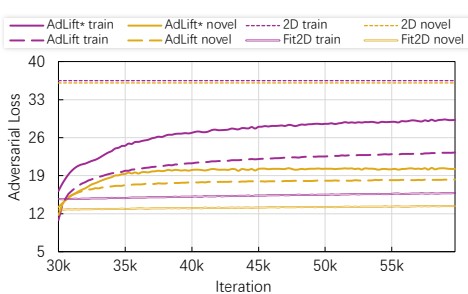

*Figure 10.* Training dynamics of Fit2D, AdLift, and AdLift* with VAE-untargeted loss $\mathcal{L}_{\text{VU}}$.

## B. Soft Constraints on Gaussian Parameters

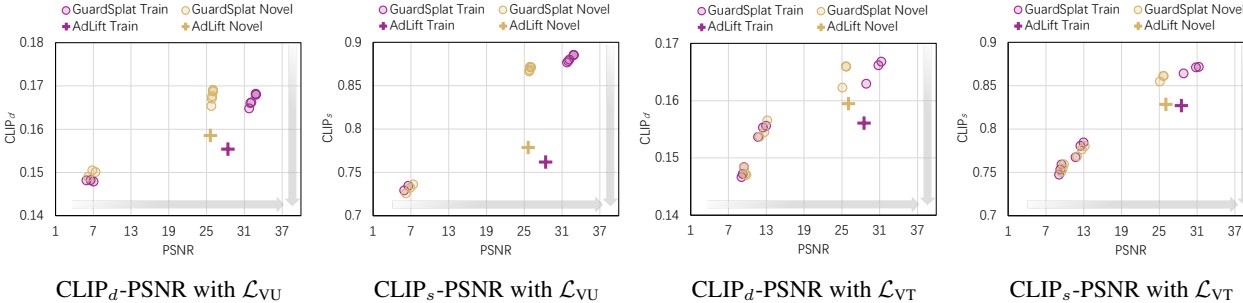

CLIP$_d$-PSNR with $\mathcal{L}_{\text{VU}}$     CLIP$_s$-PSNR with $\mathcal{L}_{\text{VU}}$     CLIP$_d$-PSNR with $\mathcal{L}_{\text{VT}}$     CLIP$_s$-PSNR with $\mathcal{L}_{\text{VT}}$

*Figure 11.* Comparison of balance between imperceptibility and protection capability. We show CLIP$_d$-PSNR scatter plot and CLIP$_s$-PSNR scatter plot on both GuardSplat and AdLift trained by VAE-targeted loss $\mathcal{L}_{\text{VT}}$ and VAE-untargeted loss $\mathcal{L}_{\text{VU}}$. (Lower CLIP$_d$/CLIP$_s$ indicates stronger protection, while higher PSNR reflects better imperceptibility).

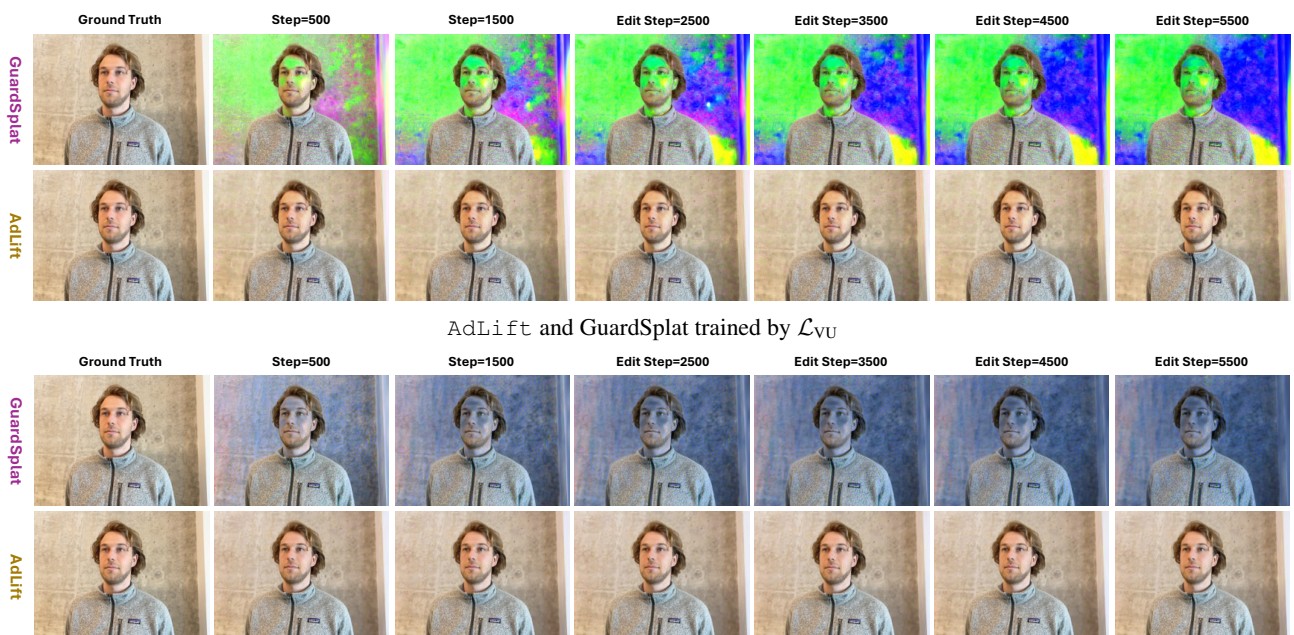

*Figure 12.* Step-wise comparison of balance between imperceptibility and protection capability. We show visualization results of our AdLift and GuardSplat (Chen et al., 2025) trained by VAE-targeted loss $\mathcal{L}_{VT}$ and VAE-untargeted loss $\mathcal{L}_{VU}$.

**Comparison with GuardSplat under different training hyperparameters.** We compare AdLift with GuardSplat (Chen et al., 2025), which perturbs the spherical-harmonic (SH) features of Gaussians under **soft** invisibility regularization. From Figure 11, GuardSplat struggles to balance imperceptibility and protection *even after searching across different hyperparameter configurations*: achieving stronger protection inevitably sacrifices imperceptibility, while preserving fidelity leads to weak protection. In contrast, AdLift achieves a much better trade-off. Besides, step-wise visualizations in Figure 12 further confirm this: GuardSplat accumulates severe color distortions due to perturbed SH features, whereas AdLift preserves natural appearance while still suppressing instruction-driven edits. These results verify that soft constraints on SH features cannot reach a satisfactory balance, underscoring the necessity of our strictly bounded constraints.

**Comparison with GuardSplat at matched PSNR or protection levels.** We further provide both quantitative and qualitative comparisons between AdLift and GuardSplat under matched PSNR or protection levels. The quantitative results are reported in Tables 4 and 5, which include: (i) comparisons under similar PSNR levels (GuardSplat-VU1 vs. AdLift-VU), and (ii) comparisons under similar anti-editing performance with matched perceptual quality (GuardSplat-VU2 vs. AdLift-VU). We also report the adversarial loss values along with the additional three editing metrics (FID (Heusel et al., 2017), $F_{1/8}$, and $F_8$ (Sajjadi et al., 2018)). Qualitative editing results comparing AdLift and GuardSplat at the similar PSNR levels are shown in Figures 13 and 14.

*Table 4.* Comparison with GuardSplat at the similar PSNR-level or anti-editing-level (Face).

| Method | SSIM ($\uparrow$) | PSNR ($\uparrow$) | LPIPS ($\downarrow$) | CLIP$_d$ ($\downarrow$) | CLIP$_s$ ($\downarrow$) | AdvLoss ($\uparrow$) | FID ($\uparrow$) | $F_{1/8}$ ($\downarrow$) | $F_8$ ($\downarrow$) |
|---|---|---|---|---|---|---|---|---|---|
| | | | | Training views | | | | | |
| No Protection | 0.9381 | 32.6436 | 0.0668 | 0.1685 | 0.8852 | – | 22.0270 | 0.9958 | 0.9947 |
| GuardSplat-VU1 | 0.9348 | 31.7765 | 0.0802 | 0.1649 | 0.8769 | 5.4381 | 26.2044 | 0.9945 | 0.9952 |
| GuardSplat-VU2 | 0.4855 | 7.0444 | 0.6414 | 0.1479 | 0.7387 | 51.6294 | 126.1376 | 0.7694 | 0.6478 |
| AdLift*-VU | 0.7001 | 28.3963 | 0.3774 | 0.1554 | 0.7620 | 29.2730 | 88.5865 | 0.8954 | 0.8742 |
| | | | | Novel views | | | | | |
| No Protection | 0.8590 | 26.0390 | 0.1292 | 0.1704 | 0.8730 | – | 76.2210 | 0.9882 | 0.9835 |
| GuardSplat-VU1 | 0.8552 | 25.7744 | 0.1442 | 0.1654 | 0.8666 | 6.3217 | 83.8178 | 0.9846 | 0.9875 |
| GuardSplat-VU2 | 0.4328 | 7.3565 | 0.6576 | 0.1501 | 0.7361 | 47.7983 | 176.4602 | 0.7253 | 0.5904 |
| AdLift*-VU | 0.6367 | 25.6115 | 0.4018 | 0.1586 | 0.7786 | 20.2232 | 129.8172 | 0.8807 | 0.9279 |

Overall, the quantitative and qualitative results consistently confirm that AdLift better balances imperceptibility and protection effectiveness: under matched PSNR levels AdLift delivers stronger editing resistance, and under matched protection strength it maintains higher perceptual quality compared to GuardSplat-based baselines.

*Table 5.* Comparison with GuardSplat at the same PSNR-level or anti-editing-level (Fangzhou).

| Method | SSIM (↑) | PSNR (↑) | LPIPS (↓) | CLIP$_d$ (↓) | CLIP$_s$ (↓) | AdvLoss (↑) | FID (↑) | F$_{1/8}$ (↓) | F$_8$ (↓) |
|---|---|---|---|---|---|---|---|---|---|
| | | | | | Training views | | | | |
| No Protection | 0.9218 | 32.1021 | 0.1093 | 0.1937 | 0.9080 | – | 23.7996 | 0.9817 | 0.9795 |
| GuardSplat-VU1 | 0.8596 | 23.4021 | 0.2126 | 0.1874 | 0.8380 | 18.4520 | 53.6308 | 0.9248 | 0.9653 |
| GuardSplat-VU2 | 0.3850 | 6.4662 | 0.6600 | 0.1727 | 0.7432 | 58.1082 | 121.3719 | 0.6165 | 0.4988 |
| AdLift*-VU | 0.6706 | 28.3497 | 0.3767 | 0.1781 | 0.7900 | 29.4458 | 84.2180 | 0.8030 | 0.7749 |
| | | | | | Novel views | | | | |
| No Protection | 0.9010 | 30.2873 | 0.1161 | 0.1938 | 0.9053 | – | 51.7357 | 0.9905 | 0.9913 |
| GuardSplat-VU1 | 0.8396 | 22.8568 | 0.2201 | 0.1904 | 0.8339 | 18.0344 | 87.1272 | 0.9270 | 0.9481 |
| GuardSplat-VU2 | 0.3765 | 6.4916 | 0.6594 | 0.1722 | 0.7374 | 56.8699 | 151.8966 | 0.6272 | 0.4956 |
| AdLift*-VU | 0.6550 | 27.4526 | 0.3762 | 0.1791 | 0.8037 | 24.3330 | 102.6782 | 0.8337 | 0.8794 |

*Table 6.* Comparison with more baselines at the similar PSNR-level (training views). Bracketed values indicate changes relative to the unprotected baseline. Best results are highlighted in **bold**.

| Method | SSIM (↑) | PSNR (↑) | LPIPS (↓) | CLIP$_d$ (↓) | CLIP$_s$ (↓) | FID (↑) | F$_{1/8}$ (↓) | F$_8$ (↓) |
|---|---|---|---|---|---|---|---|---|
| | | | | Face | | | | |
| No Protection | 0.9381 | 32.6436 | 0.0668 | 0.1685 | 0.8852 | 22.0270 | 0.9958 | 0.9947 |
| GuardSplat-(SH)-VU | 0.9348 | 31.7765 | 0.0802 | 0.1649 (-0.0036) | 0.8769 (-0.0083) | 26.2044 (+4.1774) | 0.9952 (-0.0006) | 0.9949 (0.0002) |
| GuardSplat-(SH)-VT | 0.9319 | 28.8520 | 0.0930 | 0.1630 (-0.0055) | 0.8643 (-0.0209) | 28.4026 (+6.3756) | 0.9923 (-0.0035) | 0.9919 (-0.0028) |
| GuardSplat-(PC)-VU | 0.8143 | 25.2876 | 0.1557 | 0.1605 (-0.0080) | 0.8592 (-0.0260) | 32.1277 (+10.1007) | 0.9873 (-0.0085) | 0.9910 (-0.0037) |
| GuardSplat-(PC)-VT | 0.7979 | 24.4582 | 0.1779 | 0.1583 (-0.0102) | 0.8456 (-0.0396) | 37.3899 (+15.3629) | 0.9813 (-0.0145) | 0.9822 (-0.0125) |
| GuardSplat-(Full)-VU | 0.8113 | 25.1306 | 0.1580 | 0.1617 (-0.0068) | 0.8581 (-0.0271) | 32.5037 (+10.4767) | 0.9838 (-0.0120) | 0.9887 (-0.0060) |
| GuardSplat-(Full)-VT | 0.8016 | 24.6235 | 0.1775 | 0.1581 (-0.0104) | 0.8493 (-0.0359) | 36.9855 (+14.9585) | 0.9816 (-0.0142) | 0.9818 (-0.0129) |
| GaussianMarker-VU | 0.9351 | 28.8343 | 0.0802 | 0.1596 (-0.0089) | 0.8590 (-0.0262) | 32.4431 (+10.4161) | 0.9845 (-0.0113) | 0.9916 (-0.0031) |
| GaussianMarker-VT | 0.9364 | 28.8833 | 0.0720 | 0.1620 (-0.0065) | 0.8665 (-0.0187) | 26.3699 (+4.3429) | 0.9929 (-0.0029) | 0.9942 (-0.0005) |
| AdLift*-VU | 0.7001 | 28.3963 | 0.3774 | **0.1554 (-0.0131)** | **0.7620 (-0.1232)** | **88.5865 (+66.5595)** | **0.8954 (-0.1004)** | **0.8742 (-0.1205)** |
| | | | | Fangzhou | | | | |
| No Protection | 0.9218 | 32.1021 | 0.1093 | 0.1937 | 0.9080 | 23.7996 | 0.9817 | 0.9795 |
| GuardSplat-(SH)-VU | 0.8682 | 24.3325 | 0.1952 | 0.1882 (-0.0055) | 0.8482 (-0.0598) | 49.5017 (+25.7021) | 0.9419 (-0.0398) | 0.9716 (-0.0079) |
| GuardSplat-(SH)-VT | 0.8727 | 24.2598 | 0.2198 | 0.1871 (-0.0066) | 0.8588 (-0.0492) | 43.0895 (+19.2899) | 0.9325 (-0.0492) | 0.9375 (-0.0420) |
| GuardSplat-(PC)-VU | 0.8734 | 26.4272 | 0.1810 | 0.1863 (-0.0074) | 0.8811 (-0.0269) | 32.6075 (+8.8079) | 0.9646 (-0.0171) | 0.9669 (-0.0126) |
| GuardSplat-(PC)-VT | 0.8683 | 27.0370 | 0.1756 | 0.1866 (-0.0071) | 0.8804 (-0.0276) | 35.9910 (+12.1914) | 0.9138 (-0.0679) | 0.9229 (-0.0566) |
| GuardSplat-(Full)-VU | 0.8712 | 27.5267 | 0.1743 | 0.1906 (-0.0031) | 0.8896 (-0.0184) | 32.9438 (+9.1442) | 0.9501 (-0.0316) | 0.9547 (-0.0248) |
| GuardSplat-(Full)-VT | 0.8590 | 27.0691 | 0.1821 | 0.1876 (-0.0061) | 0.8778 (-0.0302) | 36.4141 (+12.6145) | 0.9598 (-0.0219) | 0.9473 (-0.0322) |
| GaussianMarker-VU | 0.9200 | 28.0355 | 0.1455 | 0.1907 (-0.0030) | 0.8858 (-0.0222) | 34.3784 (+10.5788) | 0.9695 (-0.0122) | 0.9672 (-0.0123) |
| GaussianMarker-VT | 0.8637 | 23.7295 | 0.2460 | 0.1865 (-0.0072) | 0.8296 (-0.0784) | 57.1012 (+33.3016) | 0.9015 (-0.0802) | 0.9368 (-0.0427) |
| AdLift*-VU | 0.6706 | 28.3497 | 0.3767 | **0.1781 (-0.0156)** | **0.7900 (-0.1180)** | **84.2180 (+60.4184)** | **0.8030 (-0.1787)** | **0.7749 (-0.2046)** |

*Table 7.* Comparison with more baselines at the similar PSNR-level (novel views). Bracketed values indicate changes relative to the unprotected baseline. Best results are highlighted in **bold**.

| Method | SSIM (↑) | PSNR (↑) | LPIPS (↓) | CLIP$_d$ (↓) | CLIP$_s$ (↓) | FID (↑) | F$_{1/8}$ (↓) | F$_8$ (↓) |
|---|---|---|---|---|---|---|---|---|
| | | | | Face | | | | |
| No Protection | 0.8590 | 26.0390 | 0.1292 | 0.1704 | 0.8730 | 76.2210 | 0.9882 | 0.9835 |
| GuardSplat-(SH)-VU | 0.8552 | 25.7744 | 0.1442 | 0.1654 (-0.0050) | 0.8666 (-0.0064) | 83.8178 (+7.5968) | 0.9834 (-0.0048) | 0.9843 (0.0008) |
| GuardSplat-(SH)-VT | 0.8528 | 25.0700 | 0.1571 | 0.1623 (-0.0081) | 0.8550 (-0.0180) | 80.3380 (+4.1170) | 0.9901 (0.0019) | 0.9879 (0.0044) |
| GuardSplat-(PC)-VU | 0.7852 | 23.8763 | 0.1799 | 0.1667 (-0.0037) | 0.8575 (-0.0155) | 85.5826 (+9.3616) | 0.9789 (-0.0093) | 0.9790 (-0.0045) |
| GuardSplat-(PC)-VT | 0.7712 | 23.2702 | 0.2004 | 0.1652 (-0.0052) | 0.8480 (-0.0250) | 91.1785 (+14.9575) | 0.9752 (-0.0130) | 0.9783 (-0.0052) |
| GuardSplat-(Full)-VU | 0.7819 | 23.7877 | 0.1826 | 0.1658 (-0.0046) | 0.8535 (-0.0195) | 87.5892 (+11.3682) | 0.9801 (-0.0081) | 0.9762 (-0.0073) |
| GuardSplat-(Full)-VT | 0.7752 | 23.4051 | 0.1983 | 0.1622 (-0.0082) | 0.8505 (-0.0225) | 90.4853 (+14.2643) | 0.9748 (-0.0134) | 0.9774 (-0.0061) |
| GaussianMarker-VU | 0.8566 | 24.4911 | 0.1404 | 0.1628 (-0.0076) | 0.8578 (-0.0152) | 81.2042 (+4.9832) | 0.9772 (-0.0110) | 0.9770 (-0.0065) |
| GaussianMarker-VT | 0.8567 | 24.8557 | 0.1346 | 0.1647 (-0.0057) | 0.8608 (-0.0122) | 81.1457 (+4.9247) | 0.9849 (-0.0033) | 0.9798 (-0.0037) |
| AdLift*-VU | 0.6367 | 25.6115 | 0.4018 | **0.1586 (-0.0118)** | **0.7786 (-0.0944)** | **129.8172 (+53.5962)** | **0.8807 (-0.1075)** | **0.9279 (-0.0556)** |
| | | | | Fangzhou | | | | |
| No Protection | 0.9010 | 30.2873 | 0.1161 | 0.1938 | 0.9053 | 51.7357 | 0.9905 | 0.9913 |
| GuardSplat-(SH)-VU | 0.8484 | 23.7358 | 0.2038 | 0.1905 (-0.0033) | 0.8464 (-0.0589) | 80.7992 (+29.0635) | 0.9493 (-0.0412) | 0.9679 (-0.0234) |
| GuardSplat-(SH)-VT | 0.8546 | 23.6860 | 0.2254 | 0.1876 (-0.0062) | 0.8570 (-0.0483) | 71.4061 (+19.6704) | 0.9533 (-0.0372) | 0.9449 (-0.0464) |
| GuardSplat-(PC)-VU | 0.8608 | 25.7609 | 0.1861 | 0.1880 (-0.0058) | 0.8763 (-0.0290) | 62.2475 (+10.5118) | 0.9573 (-0.0332) | 0.9629 (-0.0284) |
| GuardSplat-(PC)-VT | 0.8570 | 26.3485 | 0.1807 | 0.1900 (-0.0038) | 0.8807 (-0.0246) | 65.9022 (+14.1665) | 0.9403 (-0.0502) | 0.9583 (-0.0330) |
| GuardSplat-(Full)-VU | 0.8582 | 26.7445 | 0.1795 | 0.1903 (-0.0035) | 0.8824 (-0.0229) | 62.6535 (+10.9178) | 0.9541 (-0.0364) | 0.9543 (-0.0370) |
| GuardSplat-(Full)-VT | 0.8480 | 26.3622 | 0.1866 | 0.1898 (-0.0040) | 0.8752 (-0.0301) | 67.569 (+15.8333) | 0.9296 (-0.0609) | 0.9403 (-0.0510) |
| GaussianMarker-VU | 0.8998 | 26.8220 | 0.1516 | 0.1914 (-0.0024) | 0.8844 (-0.0209) | 64.0244 (+12.2887) | 0.9735 (-0.0170) | 0.9726 (-0.0187) |
| GaussianMarker-VT | 0.8510 | 23.2438 | 0.2488 | 0.1872 (-0.0066) | 0.8265 (-0.0788) | 87.4162 (+35.6805) | 0.8793 (-0.1112) | 0.9319 (-0.0594) |
| AdLift*-VU | 0.6550 | 27.4526 | 0.3762 | **0.1791 (-0.0147)** | **0.8037 (-0.1016)** | **102.6782 (+50.9425)** | **0.8337 (-0.1568)** | **0.8794 (-0.1119)** |

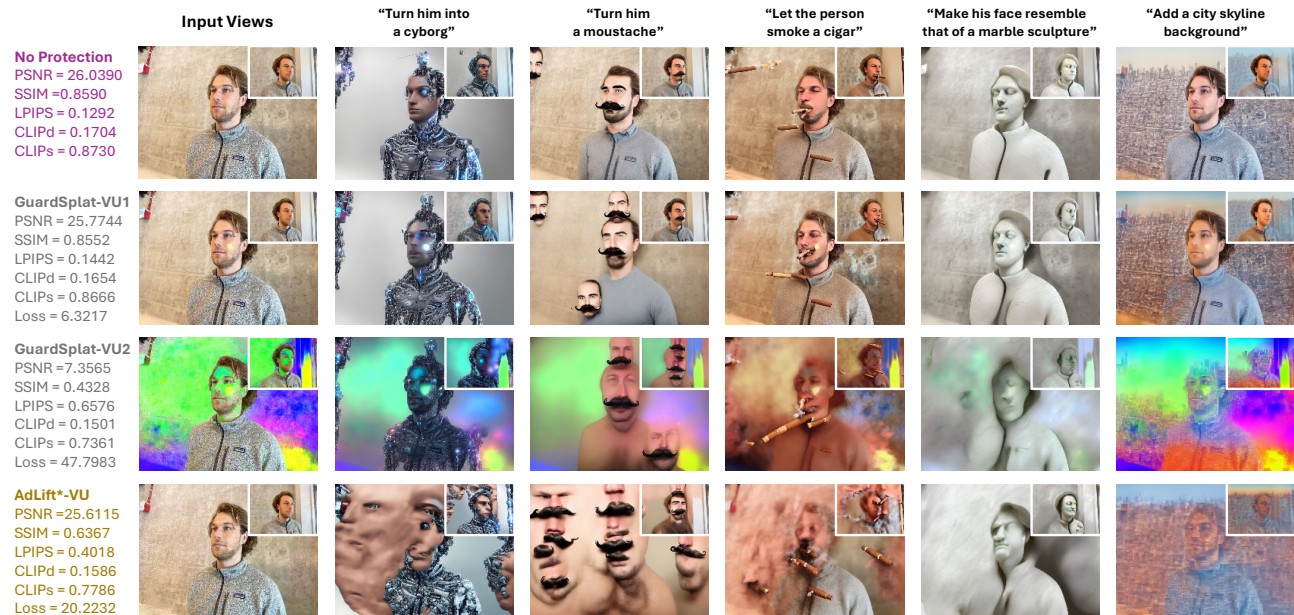

*Figure 13.* Comparison of qualitative editing results for GuardSplat-VU (with 2 different configurations) and `AdLift*`-VU at the similar PSNR-level or anti-editing-level (Face).

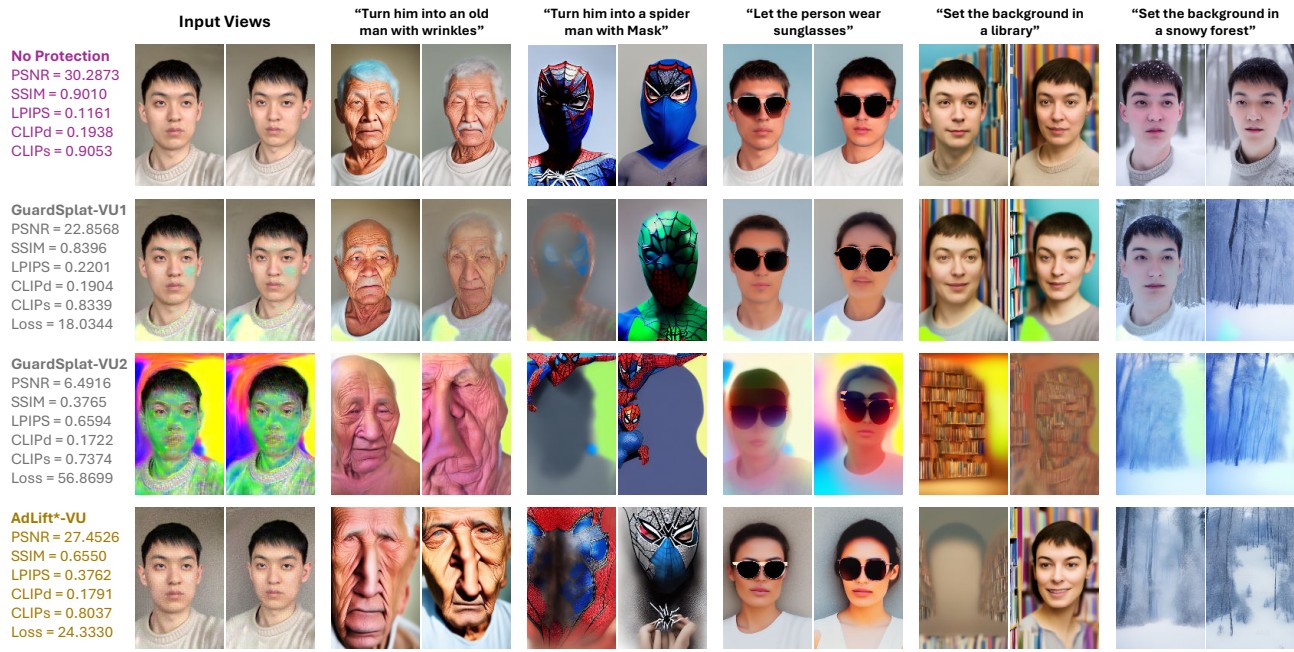

*Figure 14.* Comparison of qualitative editing results for GuardSplat-VU (with 2 different configurations) and `AdLift*`-VU at the similar PSNR-level or anti-editing-level (Fangzhou).

**Comparison with more baselines at the matched PSNR level.** We additionally include several potential baselines by adapting representative 3DGS watermarking methods into an adversarial training regime, and compare them against `AdLift` under matched perceptual quality. Specifically, we include the following variants:

- GuardSplat-(SH): Perturb only the SH features of all Gaussians, following the original GuardSplat (Chen et al., 2025).

- GuardSplat-(PC): Perturb Positions and Covariance features of all Gaussians using GuardSplat (Chen et al., 2025).

- GuardSplat-(Full): Perturb all features of all Gaussians using GuardSplat (Chen et al., 2025), including: Position, Covariance, Opacity, and SH features.

- GaussianMarker (Huang et al., 2024): Split high-uncertainty Gaussians into new Gaussians to encode perturbations.

For each method, we evaluate two adversarial objectives, untargeted VAE loss (VU) and targeted VAE loss (VT), to ensure comparison under different attack settings. We preserve each method's original fidelity-preserving loss but replace original watermark decoding loss with our adversarial objective so that all methods are optimized toward the same protection goal.

The quantitative results are provided in Table 6 (training views) and Table 7 (novel views). We report protection performance under similar perceptual quality (similar PSNR). We also include representative qualitative comparisons in Figure 15. From the results, AdLift consistently achieves stronger anti-editing performance under similar invisibility constraints.

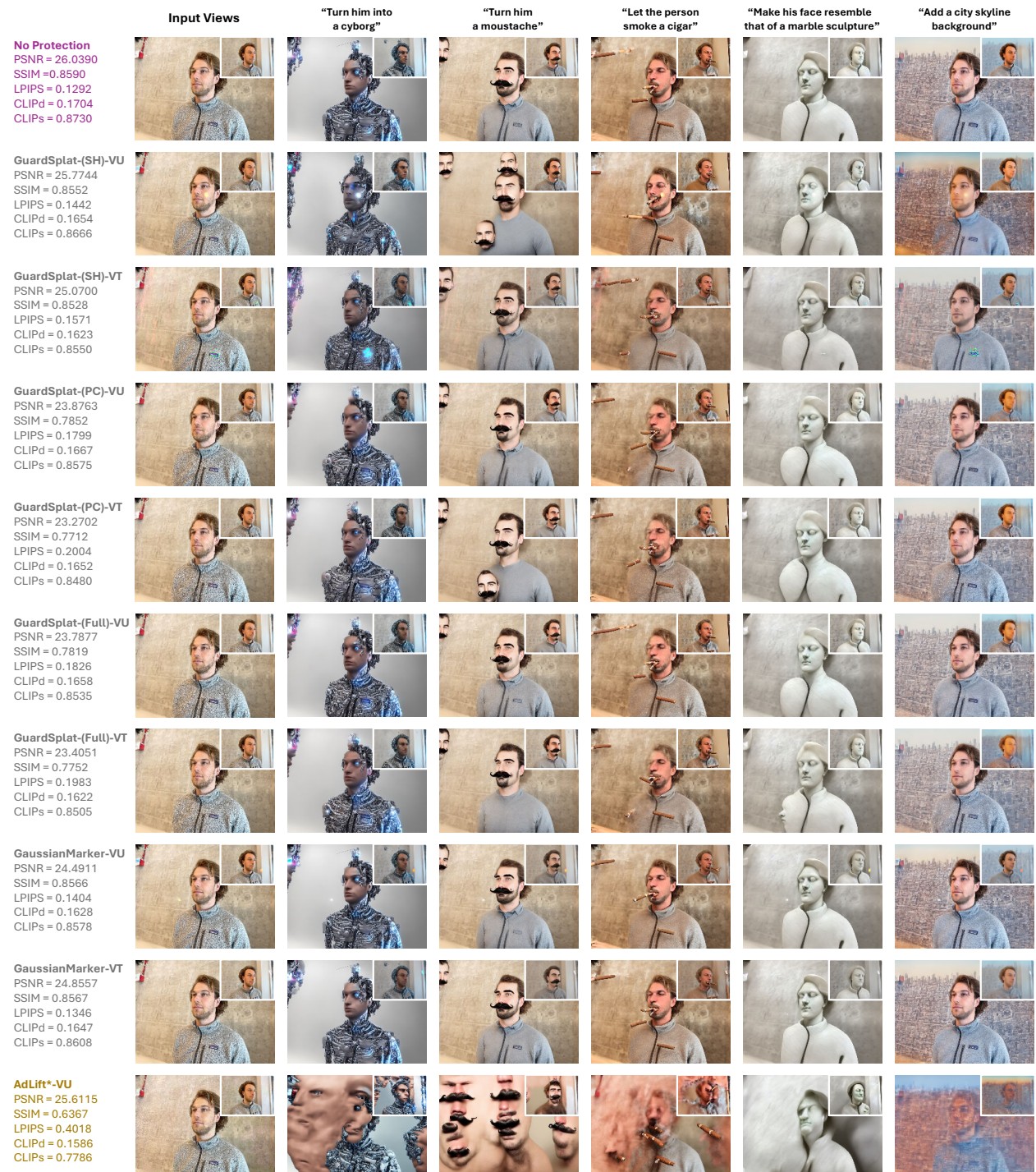

*Figure 15.* Comparison of qualitative editing results between additional baselines and AdLift*-VU (novel views).

## C. More Qualitative and Quantitative Results of **AdLift**

### C.1. Visualization of Learned Perturbations

Figure 16 shows the protection perturbations produced by `AdLift` on bear and stone-horse scenes. The perturbations remain consistent across different viewpoints, indicating strong cross-view continuity rather than view-specific noise. Moreover, the perturbations exhibit structured, semantically aligned patterns (e.g., contours along object boundaries and textures), instead of appearing as purely random pixel-level noise. Such properties highlight that the learned perturbations are not arbitrary but encode meaningful adversarial signals (Ilyas et al., 2019; Xie et al., 2019; Sabour et al., 2015) that generalize across views.

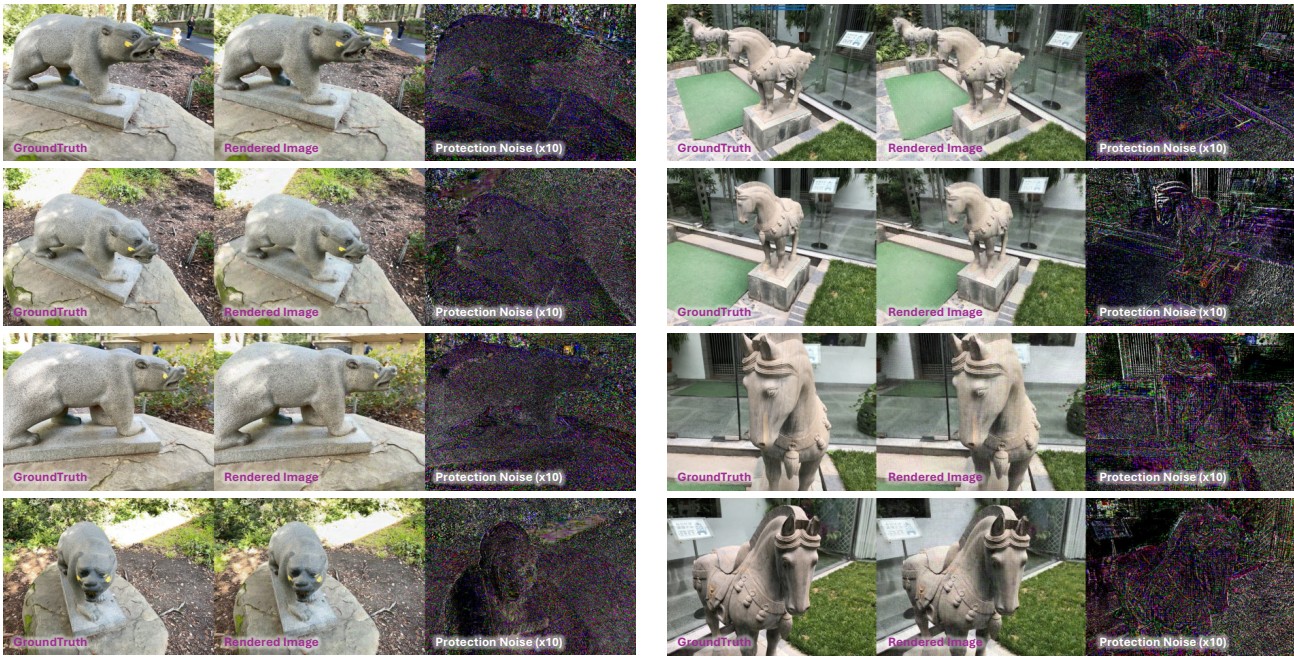

*Figure 16.* Visualization of protection perturbations learned via `AdLift*`-VU.

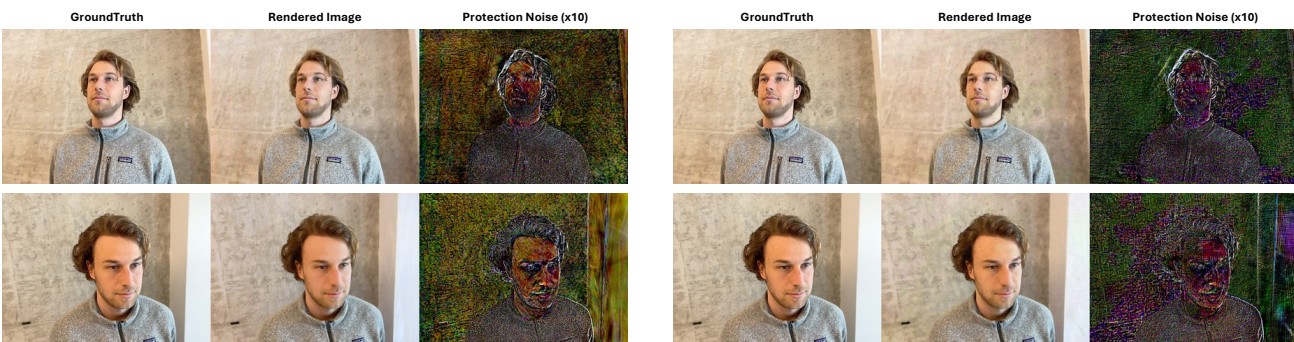

*Figure 17.* Visualization of the learned perturbations of `AdLift*` trained via different adversarial losses. (*Left*): `AdLift*` with targeted VAE loss. (*Right*): `AdLift*` with untargeted VAE loss.

### C.2. More Results on Instruction-based Image Editing

Figure 18 presents qualitative comparisons of unprotected and protected 3DGS assets under instruction-driven 2D editing with IP2P (Brooks et al., 2023). For both training and novel views, unprotected assets are highly vulnerable: IP2P faithfully executes the editing instructions, producing drastic manipulations such as adding moustaches, changing background scenes, or altering facial attributes. By contrast, assets protected with our `AdLift` exhibit strong resistance to these instruction-based edits. Across both Face and Fangzhou scenes, the protected assets fail to produce realistic editing outcomes under

*Table 8.* Quantitative evaluations on instruction-based image editing. We report results on both training and novel views. For invisibility, higher SSIM and PSNR, and lower LPIPS indicate better fidelity to the original image. For protection quality, lower $CLIP_d$ and $CLIP_s$ suggest stronger resistance to view-dependent instruction-based editing.

| Dataset | Method | Training Views | | | | | Novel Views | | | | |
|---------|--------|----------------|--|--|--|--|-------------|--|--|--|--|
| | | Invisibility | | | Protection Quality | | Invisibility | | | Protection Quality | |
| | | SSIM(↑) | PSNR(↑) | LPIPS(↓) | $CLIP_d$(↓) | $CLIP_s$(↓) | SSIM(↑) | PSNR(↑) | LPIPS(↓) | $CLIP_d$(↓) | $CLIP_s$(↓) |
| **Face** | Ground Truth | — | — | — | 0.1688 | — | — | — | — | 0.1715 | — |
| | No Protection | 0.9381 | 32.6436 | 0.0668 | 0.1685 | 0.8852 | 0.8590 | 26.0390 | 0.1292 | 0.1704 | 0.8730 |
| | Random Noise | 0.7810 | 28.3994 | 0.1785 | 0.1607 | 0.8578 | 0.7126 | 25.3380 | 0.2147 | 0.1623 | 0.8512 |
| | Fit2D-VU | 0.7446 | 28.4200 | 0.2811 | 0.1542 | 0.8192 | 0.6601 | 25.1338 | 0.3103 | 0.1564 | 0.8165 |
| | Fit2D-VT | 0.8346 | 29.5256 | 0.1547 | 0.1567 | 0.8515 | 0.7575 | 26.2201 | 0.1960 | 0.1591 | 0.8532 |
| | AdLift-VU | 0.7488 | 28.6118 | 0.3308 | 0.1561 | 0.7990 | 0.6764 | 24.4293 | 0.3718 | 0.1572 | 0.7998 |
| | AdLift-VT | 0.8075 | 28.6309 | 0.2011 | 0.1574 | 0.8326 | 0.7340 | 24.5285 | 0.2552 | 0.1591 | 0.8272 |
| | AdLift*-VU | 0.7001 | 28.3963 | 0.3774 | 0.1554 | 0.7620 | 0.6367 | 25.6115 | 0.4018 | 0.1586 | 0.7786 |
| | AdLift*-VT | 0.7842 | 28.5198 | 0.2250 | 0.1561 | 0.8272 | 0.7239 | 26.0586 | 0.2522 | 0.1595 | 0.8285 |
| **Fang-zhou** | Ground Truth | — | — | — | 0.2016 | — | — | — | — | 0.2004 | — |
| | No Protection | 0.9218 | 32.1021 | 0.1093 | 0.1937 | 0.9080 | 0.9010 | 30.2873 | 0.1161 | 0.1938 | 0.9053 |
| | Random Noise | 0.7073 | 27.9112 | 0.1995 | 0.1955 | 0.8899 | 0.6849 | 26.9362 | 0.2107 | 0.1944 | 0.8867 |
| | Fit2D-VU | 0.7164 | 28.4063 | 0.2976 | 0.1919 | 0.8595 | 0.6853 | 27.3809 | 0.3116 | 0.1937 | 0.8952 |
| | Fit2D-VT | 0.8415 | 29.6154 | 0.1442 | 0.1883 | 0.8801 | 0.8163 | 28.3239 | 0.1574 | 0.1890 | 0.8796 |
| | AdLift-VU | 0.7386 | 28.5668 | 0.2783 | 0.1770 | 0.8090 | 0.7245 | 27.6556 | 0.2829 | 0.1783 | 0.8092 |
| | AdLift-VT | 0.8142 | 28.6419 | 0.1916 | 0.1792 | 0.8314 | 0.8021 | 27.8007 | 0.1972 | 0.1805 | 0.8290 |
| | AdLift*-VU | 0.6706 | 28.3497 | 0.3767 | 0.1781 | 0.7900 | 0.6550 | 27.4526 | 0.3762 | 0.1791 | 0.8037 |
| | AdLift*-VT | 0.8009 | 28.6429 | 0.2156 | 0.1776 | 0.8198 | 0.7903 | 27.6700 | 0.2205 | 0.1788 | 0.8148 |
| **Bear** | Ground Truth | — | — | — | 0.1858 | — | — | — | — | 0.1863 | — |
| | No Protection | 0.9111 | 27.0824 | 0.0885 | 0.1852 | 0.8734 | 0.7623 | 21.2934 | 0.1800 | 0.1856 | 0.8626 |
| | Random Noise | 0.8048 | 24.8107 | 0.1545 | 0.1806 | 0.8575 | 0.6793 | 20.4670 | 0.2342 | 0.1836 | 0.8527 |
| | Fit2D-VU | 0.8241 | 25.0807 | 0.2042 | 0.1785 | 0.8407 | 0.6844 | 20.5878 | 0.2792 | 0.1786 | 0.8394 |
| | Fit2D-VT | 0.8349 | 25.4106 | 0.1433 | 0.1718 | 0.8411 | 0.6916 | 20.7411 | 0.2281 | 0.1726 | 0.8361 |
| | AdLift-VU | 0.8401 | 26.1373 | 0.1835 | 0.1849 | 0.8373 | 0.7100 | 21.0489 | 0.2595 | 0.1851 | 0.8336 |
| | AdLift-VT | 0.8353 | 26.1047 | 0.1402 | 0.1777 | 0.8402 | 0.7061 | 21.0953 | 0.2201 | 0.1774 | 0.8342 |
| | AdLift*-VU | 0.8081 | 25.7041 | 0.2145 | 0.1865 | 0.8232 | 0.6755 | 20.6248 | 0.2924 | 0.1833 | 0.8306 |
| | AdLift*-VT | 0.8092 | 25.8351 | 0.1541 | 0.1769 | 0.8291 | 0.6782 | 20.7417 | 0.2401 | 0.1723 | 0.8253 |
| **Horse** | Ground Truth | — | — | — | 0.2043 | — | — | — | — | 0.2138 | — |
| | No Protection | 0.9149 | 30.0968 | 0.0575 | 0.2025 | 0.8903 | 0.7019 | 22.5986 | 0.1033 | 0.2105 | 0.8865 |
| | Random Noise | 0.7055 | 25.7536 | 0.1581 | 0.1944 | 0.8694 | 0.5492 | 21.4967 | 0.1882 | 0.2050 | 0.8660 |
| | Fit2D-VU | 0.7375 | 26.3795 | 0.2433 | 0.1941 | 0.8276 | 0.5615 | 21.7023 | 0.2515 | 0.2014 | 0.8439 |
| | Fit2D-VT | 0.7817 | 27.1458 | 0.1311 | 0.1838 | 0.8538 | 0.6016 | 21.9937 | 0.1660 | 0.1984 | 0.8679 |
| | AdLift-VU | 0.7730 | 27.6010 | 0.2185 | 0.1967 | 0.8102 | 0.6097 | 22.0334 | 0.2223 | 0.1989 | 0.8297 |
| | AdLift-VT | 0.7963 | 27.8463 | 0.1352 | 0.1848 | 0.8439 | 0.6305 | 22.1679 | 0.1656 | 0.1959 | 0.8560 |
| | AdLift*-VU | 0.7332 | 27.1573 | 0.2589 | 0.2023 | 0.7804 | 0.5822 | 21.9479 | 0.2615 | 0.1996 | 0.8217 |
| | AdLift*-VT | 0.7642 | 27.4353 | 0.1445 | 0.1819 | 0.8271 | 0.6106 | 21.8557 | 0.1830 | 0.1927 | 0.8496 |

given instructions. While unprotected assets can be directly edited to follow instructions with faithful preservation of facial structures, the protected results instead appear unnatural and deviate from photorealism, effectively preventing meaningful semantic modifications. Notably, this protection generalizes from training views to novel views, confirming that our lifted perturbations enforce view-consistent protection rather than overfitting to the supervised views. These results validate that `AdLift` provides effective and generalizable safeguards against instruction-guided editing attacks.

Beside, the full quantitative results on each scenes are shown in Table 8. In details, random noise fails to provide effective defense, as edits are still successfully applied. Fit2D variants partially suppress editing, but the protection is view-inconsistent and often leaves residual manipulations in novel views. In contrast, our `AdLift` variants achieve much stronger and more stable protection.

## C.3. More Results on Instruction-based 3DGS Editing

The full quantitative results of instruction-based 3DGS global editing and local editing are shown in Appendix C.3 and Table 10, respectively. In addition, we show more qualitative results in Figure 19. Our `AdLift` disrupts both global and local instruction-driven 3DGS edits, either by preventing intended transformations or by misguiding region localization to preserve the original appearance.

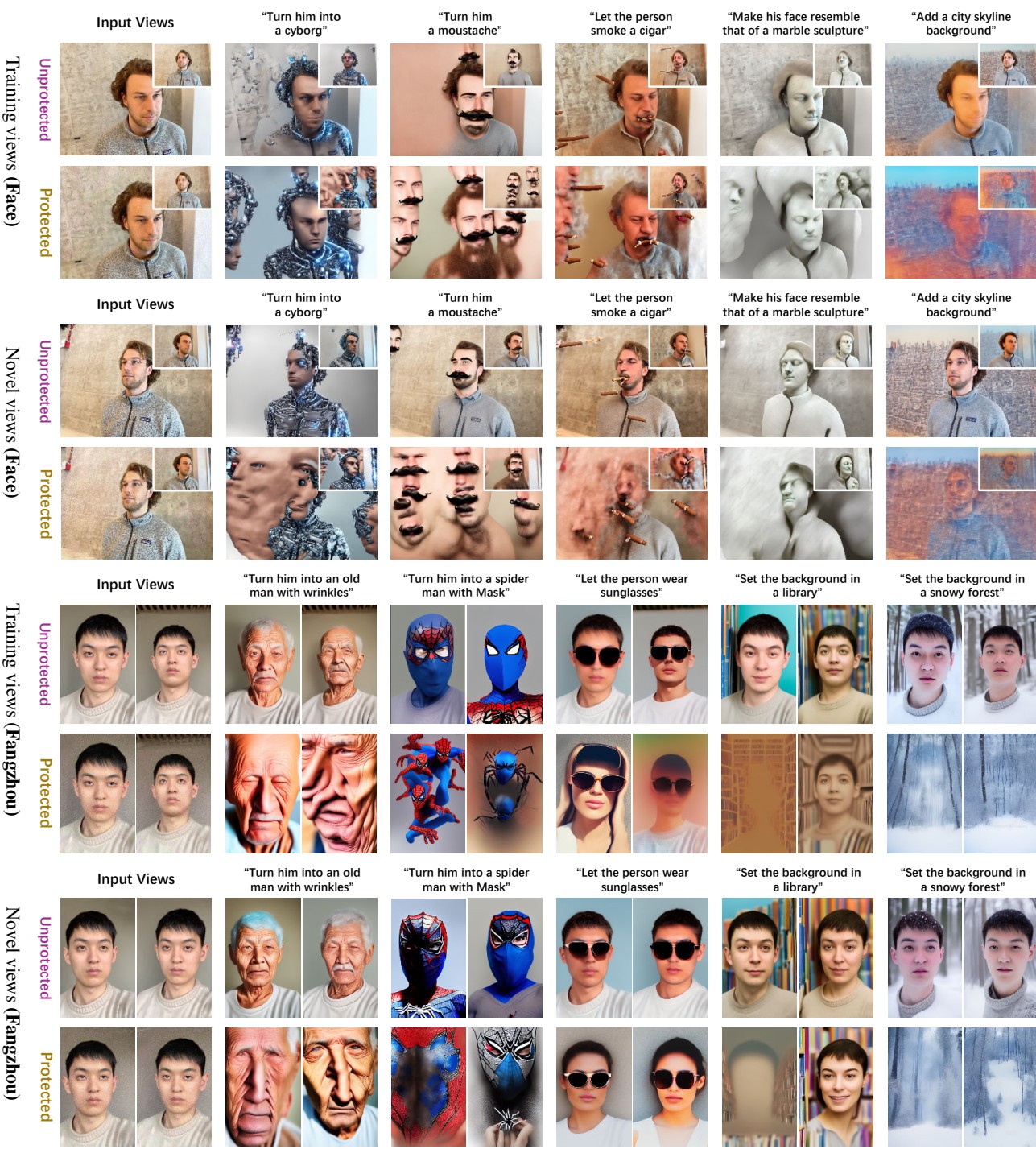

*Figure 18.* Qualitative results of instruction-based image editing on face scenes. We show the comparison of editing results of unprotected 3D assets and 3D assets protected by `AdLift`*-VU.

*Table 9.* Quantitative evaluations on instruction-based global 3DGS editing. We report results on both training and novel views.

| Dataset | Method | Instruct-GS2GS | | | | DGE (Global) | | | |
| | | Training Views | | Novel Views | | Training Views | | Novel Views | |
| | | CLIP$_d$($\downarrow$) | CLIP$_s$($\downarrow$) | CLIP$_d$($\downarrow$) | CLIP$_s$($\downarrow$) | CLIP$_d$($\downarrow$) | CLIP$_s$($\downarrow$) | CLIP$_d$($\downarrow$) | CLIP$_s$($\downarrow$) |
|---|---|---|---|---|---|---|---|---|---|
| **Face** | No Protection | 0.1239 | — | 0.1108 | — | 0.1240 | — | 0.1218 | — |
| | `AdLift*`-VT | 0.1138 | 0.8758 | 0.1031 | 0.8682 | 0.1162 | 0.9300 | 0.1166 | 0.9298 |
| | `AdLift*`-VU | 0.1107 | 0.7914 | 0.0857 | 0.8046 | 0.1154 | 0.8986 | 0.1200 | 0.9029 |
| **Fangzhou** | No Protection | 0.1339 | — | 0.1324 | — | 0.1710 | — | 0.1713 | — |
| | `AdLift*`-VT | 0.1151 | 0.8734 | 0.1119 | 0.87305 | 0.1431 | 0.8877 | 0.1432 | 0.8875 |
| | `AdLift*`-VU | 0.1194 | 0.8269 | 0.1160 | 0.8259 | 0.1523 | 0.8633 | 0.1522 | 0.8630 |
| **Bear** | No Protection | 0.1351 | — | 0.1273 | — | 0.1025 | — | 0.1009 | — |
| | `AdLift*`-VT | 0.1484 | 0.8483 | 0.1292 | 0.8529 | 0.0901 | 0.9398 | 0.0890 | 0.9400 |
| | `AdLift*`-VU | 0.1386 | 0.8886 | 0.1236 | 0.8951 | 0.0911 | 0.9318 | 0.0897 | 0.9308 |
| **Horse** | No Protection | 0.1557 | — | 0.1300 | — | 0.1174 | — | 0.1243 | — |
| | `AdLift*`-VT | 0.1512 | 0.8683 | 0.1223 | 0.8831 | 0.1088 | 0.9301 | 0.1160 | 0.9341 |
| | `AdLift*`-VU | 0.1806 | 0.8033 | 0.1468 | 0.8029 | 0.1037 | 0.9157 | 0.1086 | 0.9179 |

*Table 10.* Quantitative evaluations on instruction-based 3DGS local editing

| Dataset | Method | DGE (Local) | | | |
| | | Training Views | | Novel Views | |
| | | CLIP$_d$($\downarrow$) | CLIP$_s$($\downarrow$) | CLIP$_d$($\downarrow$) | CLIP$_s$($\downarrow$) |
|---|---|---|---|---|---|
| **Face** | No Protection | 0.1375 | — | 0.1320 | — |
| | `AdLift`-ST | 0.0011 | 0.8347 | 0.0207 | 0.8473 |
| **Fangzhou** | No Protection | 0.2000 | — | 0.1951 | — |
| | `AdLift`-ST | 0.0297 | 0.8016 | 0.0236 | 0.7989 |
| **Bear** | No Protection | 0.0946 | — | 0.0979 | — |
| | `AdLift`-ST | -0.0038 | 0.8676 | 0.0024 | 0.8610 |
| **Horse** | No Protection | 0.1024 | — | 0.1022 | — |
| | `AdLift`-ST (0.8) | 0.0723 | 0.9073 | 0.0733 | 0.9173 |

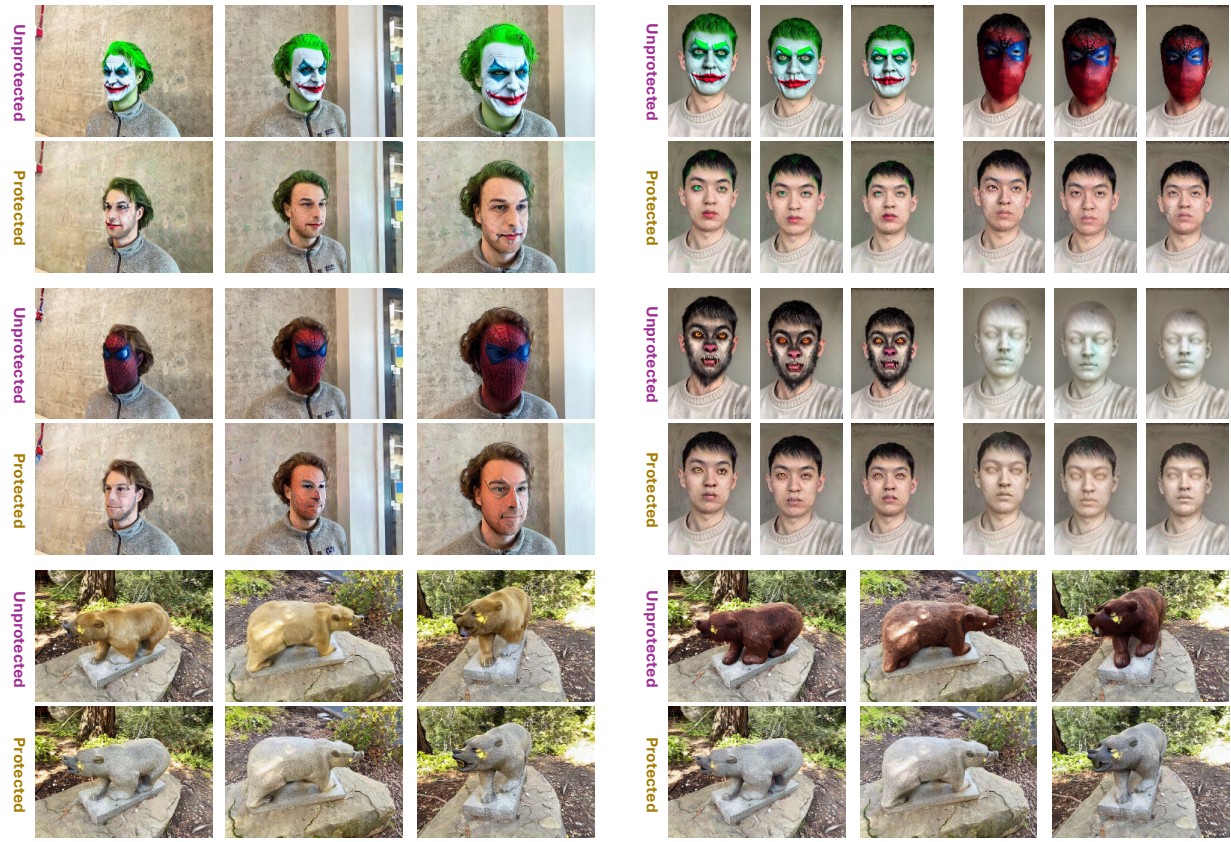

*Figure 19.* Instruction-based local 3DGS editing for unprotected and `AdLift`*-ST protected assets.

# D. Robustness to Editing Hyperparameters

An important requirement for practical protection is robustness against variations in editing hyperparameters, since adversaries can freely adjust parameters of instruction-driven editing models. To evaluate this, we evaluate our `AdLift` under different settings of `guidance_scale`, `image_guidance_scale`, and `num_inference_steps`.

**Guidance Scale.** As shown in Figure 20, increasing the `guidance_scale` generally strengthens the editing effect on unprotected assets, leading to higher $CLIP_d$ and $CLIP_s$ scores. In contrast, our protected assets maintain consistently low values for both CLIP scores across a wide range of values. This indicates that `AdLift` provides stable protection even when the editing strength is increased.

**Image Guidance Scale.** As shown in Figure 21, our protection demonstrates superiority, consistently achieving lower $CLIP_d$ and $CLIP_s$ scores and thus stronger resistance for a wide range of `image_guidance_scale`. However, extreme values of `image_guidance_scale` (either too small or too large) render the editing itself ineffective.

**Number of Inference Steps.** Figure 22 shows results under different numbers of diffusion inference steps. Although both the $CLIP_d$ and $CLIP_s$ scores increase as the number of steps increases, the gap compared to unprotected assets remains substantial, illustrating the effectiveness of `AdLift` under varying editing steps.

In addition, we provide visualizations under different editing hyperparameters in Figures 23 to 25, which further verify the effectiveness and stability of our method across diverse editing configurations. Overall, across all tested hyperparameters, `AdLift` consistently outperforms baselines, yielding lower $CLIP_d$ and $CLIP_s$ on both training and novel views. These results highlight that our protection is not tied to specific editing settings, but instead provides strong and generalizable robustness against diverse editing configurations.

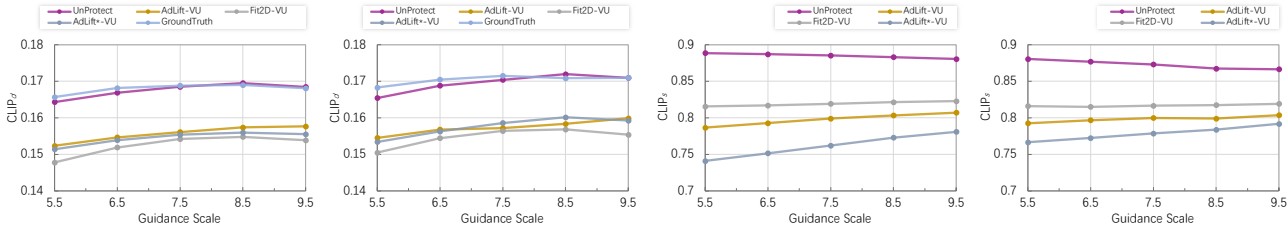

| $CLIP_d$ of training views | $CLIP_d$ of novel views | $CLIP_s$ of training views | $CLIP_s$ of novel views |

*Figure 20.* The robustness of `AdLift` against different **guidance_scale** in *instruction-based image editing* (fixing `num_inference_steps=10`, `image_guidance_scale=1.0`). Smaller $CLIP_d$ and $CLIP_s$ indicate stronger protection.

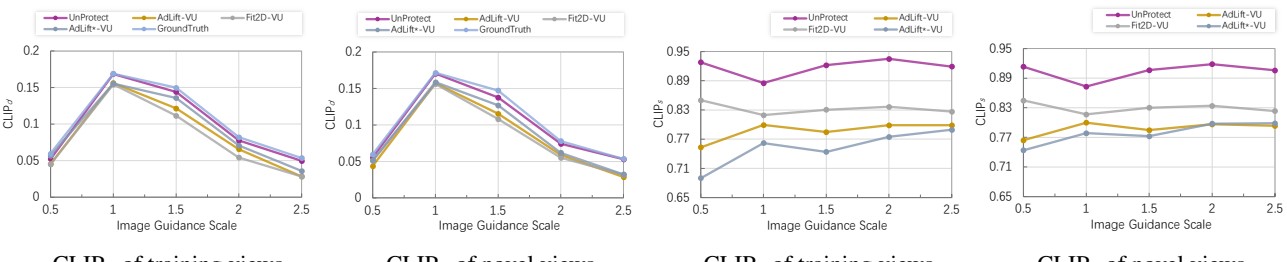

| $CLIP_d$ of training views | $CLIP_d$ of novel views | $CLIP_s$ of training views | $CLIP_s$ of novel views |

*Figure 21.* The robustness of `AdLift` against different **image_guidance_scale** in *instruction-based image editing* (fixing `num_inference_steps=10`, `guidance_scale=7.5`). Smaller $CLIP_d$ and $CLIP_s$ indicate stronger protection.

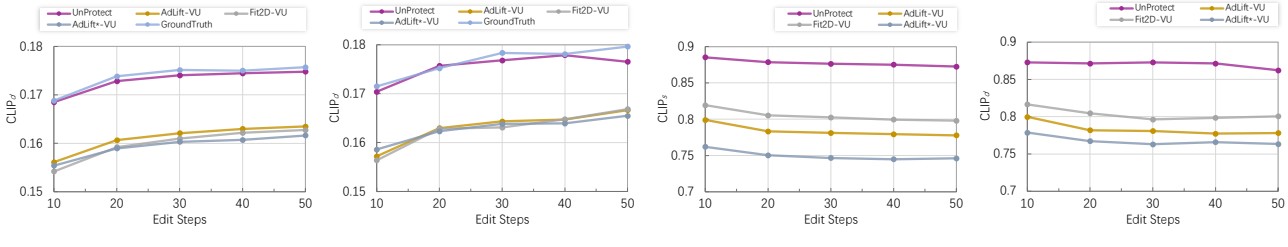

| $CLIP_d$ of training views | $CLIP_d$ of novel views | $CLIP_s$ of training views | $CLIP_s$ of novel views |

*Figure 22.* The robustness of `AdLift` against different **num_inference_steps** in *instruction-based image editing* (fixing `image_guidance_scale=1.0`, `guidance_scale=7.5`). Smaller $CLIP_d$ and $CLIP_s$ indicate stronger protection.

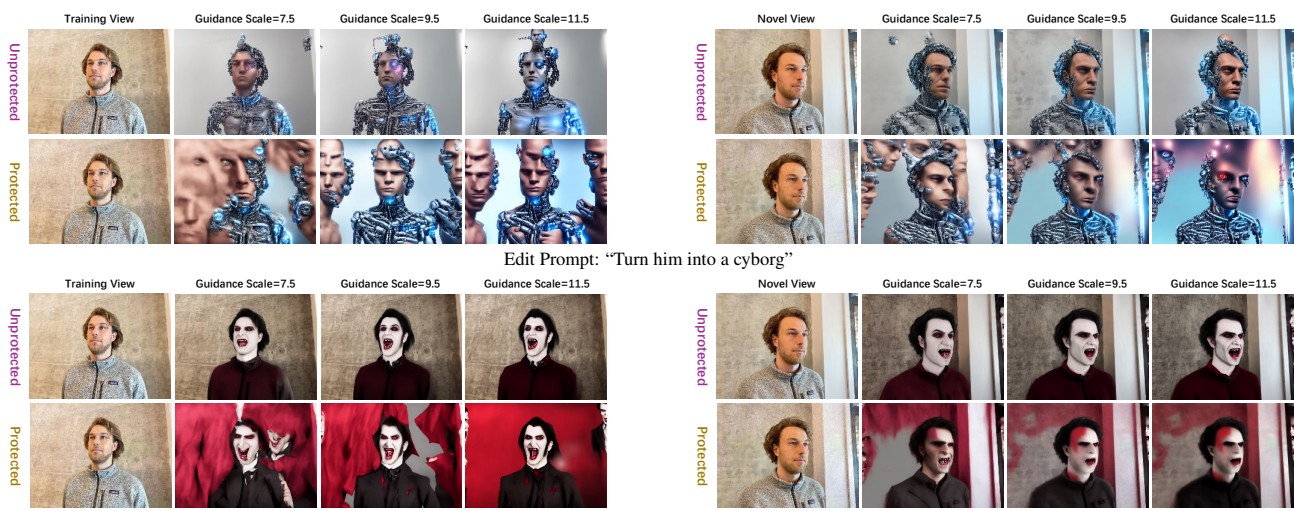

*Figure 23.* Visualization of editing under different `guidance_scale` setting (unprotected 3DGS assets vs. protected by `AdLift*`-VU).

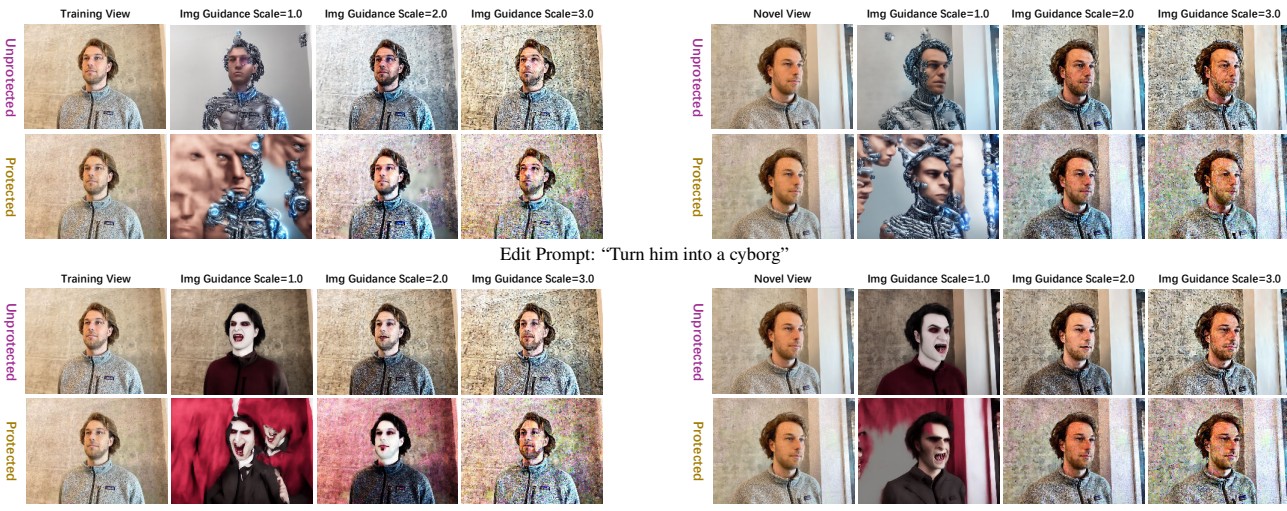

*Figure 24.* Visualization of editing under different `image_guidance_scale` setting (unprotected vs. protected by `AdLift*`-VU).

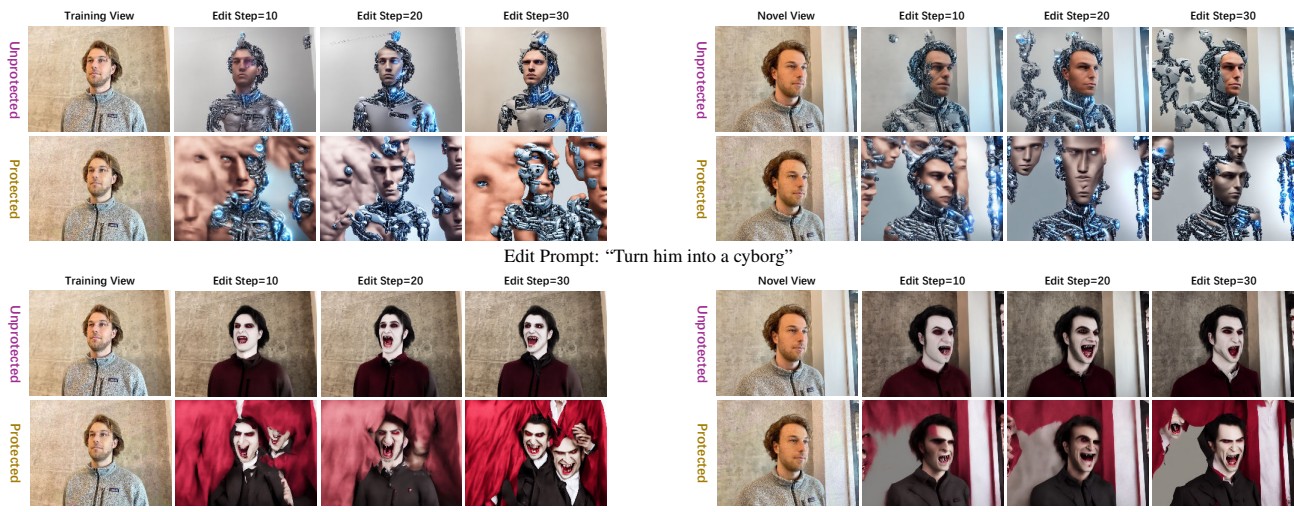

*Figure 25.* Visualization of editing under different `num_inference_steps` setting (unprotected vs. protected by `AdLift*`-VU).

# E. Analysis of Different Protection Hyperparameters

Figures 26 to 29 analyze the influence of different training hyperparameters for `AdLift`, including the perturbation budget $\eta$, learning rate $\alpha$, gradient truncation steps $K_p$, and image-to-Gaussian fitting steps $K_l$. From the results, the perturbation budget $\eta$ plays the most critical role: larger $\eta$ significantly enhances protection strength (lower $\text{CLIP}_d$/$\text{CLIP}_s$), but at the cost of noticeable degradation in visual quality (lower SSIM/PSNR, higher LPIPS). This indicates a clear trade-off between imperceptibility and protection effectiveness, and choosing an appropriate budget is essential in practice. In contrast, other hyperparameters have only minor influence, with protection and fidelity metrics remaining largely stable across different settings.

We also show visualization results of editing protected 3DGS assets by `AdLift` with different budgets in Figure 30, which further illustrates the trade-off between imperceptibility and protection effectiveness. These qualitative comparisons are consistent with our quantitative analysis.

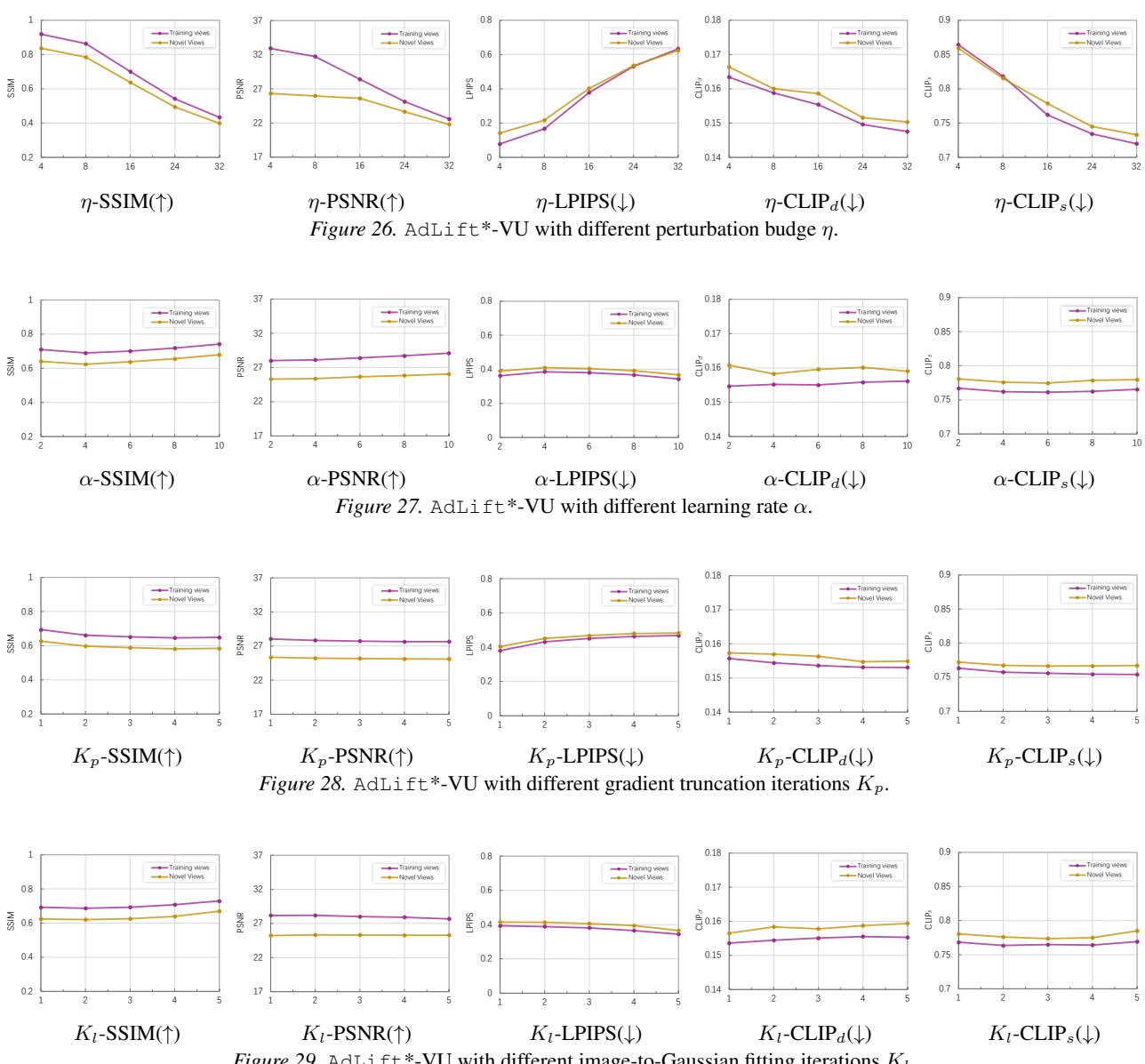

*Figure 26.* `AdLift`*-VU with different perturbation budge $\eta$.

*Figure 27.* `AdLift`*-VU with different learning rate $\alpha$.

*Figure 28.* `AdLift`*-VU with different gradient truncation iterations $K_p$.

*Figure 29.* `AdLift`*-VU with different image-to-Gaussian fitting iterations $K_l$.

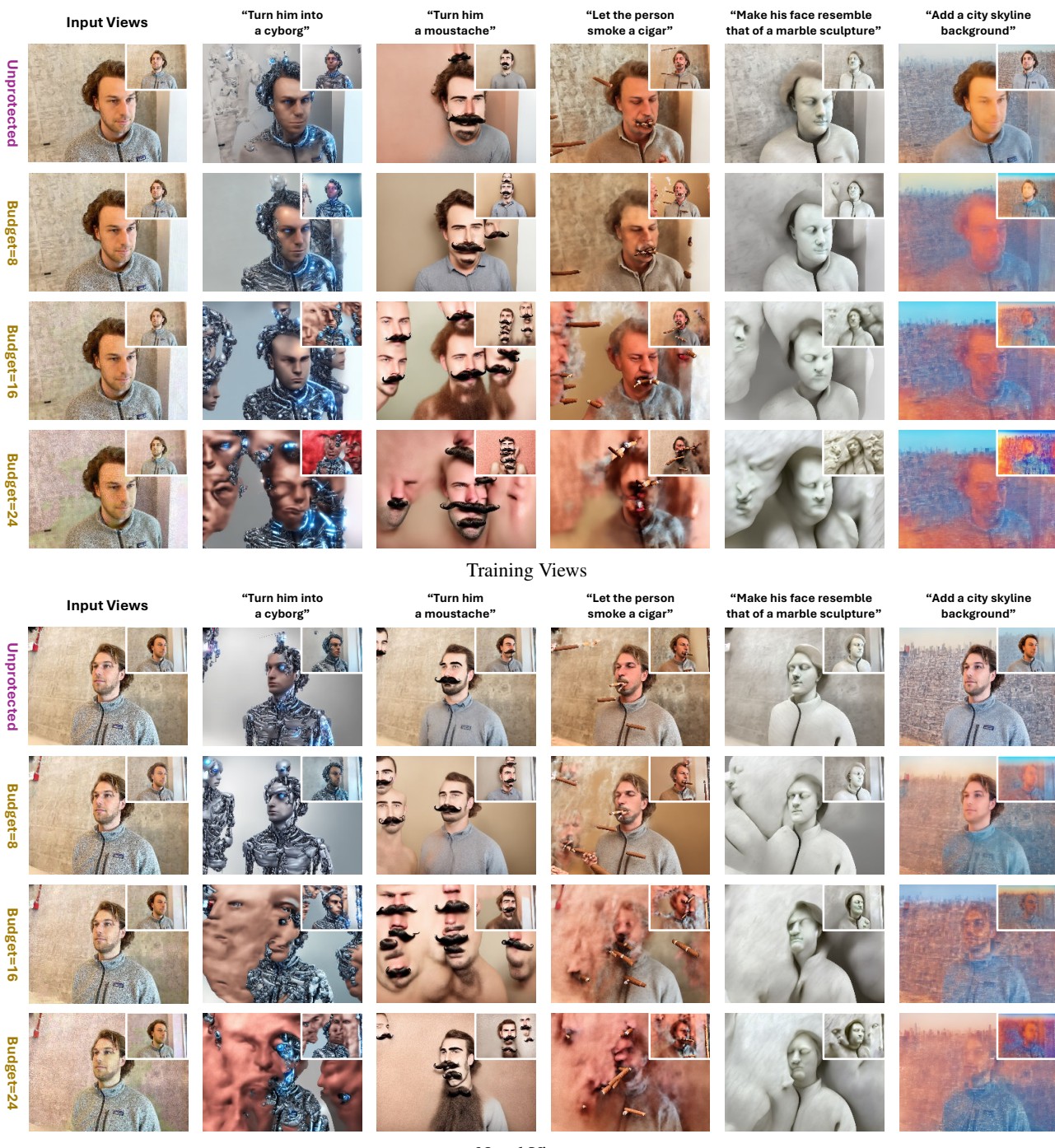

*Figure 30.* Visualization results of different perturbation budget $\eta$. We show editing results of unprotected 3DGS assets and 3DGS assets protected by `AdLift*`-VU.

# F. Transferability of **AdLift** and Robustness to Purification

## F.1. Transferability of **AdLift**

We additionally evaluate whether AdLift, which aims at lifting 2D adversarial perturbations into the 3D Gaussian space, can inherit the transferability in 2D adversarial protection.

**Experimental setup:** We train AdLift using IP2P (Brooks et al., 2023) as the surrogate editing model, and then test it against three unseen editing models, including different editing pipelines and fine-tuned variants:

- MagicBrush (Zhang et al., 2023): Enhanced fine-tuned version of SDv1.5-IP2P on MagicBrush.

- SDEdit (Meng et al., 2021): A diffusion-based editing framework that generates edited images by noising and denoising an input image, without requiring instruction-conditioned fine-tuning.

- SDXL-IP2P (Brooks et al., 2023; Podell et al., 2023): Instruction fine-tuning of Stable Diffusion XL (SDXL).

The quantitative results of transferability are reported in Table 11 (training views) and Table 12 (novel views). Qualitative results are shown in Figure 31. Empirically, across all unseen editing models, AdLift can degrade editing quality compared to the unprotected 3DGS asset, demonstrating that the protective effect generalizes beyond the surrogate model. This suggests that the proposed AdLift preserves the transferability behavior of 2D adversarial perturbations, even after being lifted into the 3D Gaussian domain.

*Table 11.* Transferability of AdLift (training views). Bracketed values indicate changes relative to the unprotected baseline. Best results are highlighted in **bold**.

| Edit Pipelines | Method | CLIP$_d$ ($\downarrow$) | CLIP$_s$ ($\downarrow$) | FID ($\uparrow$) | F$_{1/8}$ ($\downarrow$) | F$_8$ ($\downarrow$) |
|---|---|---|---|---|---|---|
| | | Face | | | | |
| IP2P | No Protection | 0.1685 | 0.8852 | 22.0270 | 0.9958 | 0.9947 |
| | AdLift*-VU | **0.1554 (-0.0131)** | **0.7620 (-0.1232)** | **88.5865 (+66.5595)** | **0.8954 (-0.1004)** | **0.8742 (-0.1205)** |
| | AdLift*-VT | 0.1561 (-0.0124) | 0.8272 (-0.0580) | 45.1351 (+23.1081) | 0.9545 (-0.0413) | 0.9650 (-0.0297) |
| MagicBrush | No Protection | 0.1741 | 0.8776 | 22.1594 | 0.9926 | 0.9885 |
| | AdLift*-VU | **0.1420 (-0.0321)** | **0.7457 (-0.1319)** | **69.4154 (+47.2560)** | **0.9305 (-0.0621)** | **0.9297 (-0.0588)** |
| | AdLift*-VT | 0.1588 (-0.0153) | 0.8138 (-0.0638) | 48.8487 (+26.6893) | 0.9642 (-0.0284) | 0.9540 (-0.0345) |
| SDEdit | No Protection | 0.1659 | 0.7811 | 25.2357 | 0.9913 | 0.9894 |
| | AdLift*-VU | **0.1448 (-0.0211)** | 0.7141 (-0.067) | 58.5985 (+33.3628) | 0.8436 (-0.1477) | **0.9484 (-0.0410)** |
| | AdLift*-VT | 0.1633 (-0.0026) | **0.7131 (-0.068)** | **62.6101 (+37.3744)** | **0.8372 (-0.1541)** | 0.9532 (-0.0362) |
| SDXL-IP2P | No Protection | 0.1053 | 0.7124 | 26.7935 | 0.9908 | 0.9899 |
| | AdLift*-VU | 0.1031 (-0.0022) | **0.6509 (-0.0615)** | **48.6225 (+21.8290)** | **0.9691 (-0.0217)** | 0.9540 (-0.0359) |
| | AdLift*-VT | **0.1016 (-0.0037)** | 0.6681 (-0.0443) | 44.9257 (+18.1322) | 0.9771 (-0.0137) | **0.9512 (-0.0387)** |
| | | Fangzhou | | | | |
| IP2P | No Protection | 0.1937 | 0.9080 | 23.7996 | 0.9817 | 0.9795 |
| | AdLift*-VU | 0.1781 (-0.0156) | **0.7900 (-0.1180)** | **84.2180 (+60.4184)** | **0.8030 (-0.1787)** | **0.7749 (-0.2046)** |
| | AdLift*-VT | **0.1776 (-0.0161)** | 0.8198 (-0.0882) | 56.5018 (+32.7022) | 0.8721 (-0.1096) | 0.8839 (-0.0956) |
| MagicBrush | No Protection | 0.1839 | 0.8583 | 40.1794 | 0.8484 | 0.7938 |
| | AdLift*-VU | 0.1625 (-0.0214) | **0.7548 (-0.1035)** | **105.8215 (+65.6421)** | 0.7135 (-0.1349) | 0.5795 (-0.2143) |
| | AdLift*-VT | **0.1611 (-0.0228)** | 0.8021 (-0.0562) | 89.8512 (+49.6718) | **0.5955 (-0.2529)** | **0.5720 (-0.2218)** |
| SDEdit | No Protection | 0.1224 | 0.7587 | 19.3086 | 0.9785 | 0.9795 |
| | AdLift*-VU | 0.1213 (-0.0011) | 0.7148 (-0.0439) | 88.8757 (+69.5671) | 0.6992 (-0.2793) | 0.8442 (-0.1353) |
| | AdLift*-VT | **0.1179 (-0.0045)** | **0.6794 (-0.0793)** | **109.4655 (+90.1569)** | **0.5921 (-0.3864)** | **0.7453 (-0.2342)** |
| SDXL-IP2P | No Protection | 0.0777 | 0.7533 | 29.4324 | 0.9422 | 0.9106 |
| | AdLift*-VU | **0.0719 (-0.0058)** | **0.7322 (-0.0211)** | **47.7706 (+18.3382)** | 0.9354 (-0.0068) | 0.8700 (-0.0406) |
| | AdLift*-VT | 0.0738 (-0.0039) | 0.7338 (-0.0195) | 44.6247 (+15.1923) | **0.9273 (-0.0149)** | **0.8019 (-0.1087)** |

## F.2. Robustness to Purification

To evaluate robustness under purification, we follow the evaluation protocol used in prior 2D anti-editing works (Liang et al., 2023; Xue et al., 2024) to test our method against two purification strategies, including a diffusion-based purification method (DiffPure (Nie et al., 2022)) and a standard compression method (JPEG compression (Sandoval-Segura et al., 2023)). The quantitative results are reported in Table 13 (training views) and Table 14 (novel views). Qualitative results are shown in Figure 32.

Our findings show that AdLift exhibits similar behavior to 2D adversarial protection: they remain effective under JPEG compression-based purification and generalize across unseen views. While diffusion-based purification reduces protection strength, AdLift still outperforms unprotected 3DGS assets.

*Table 12.* Transferability of `AdLift` (novel views). Bracketed values indicate changes relative to the unprotected baseline. Best results are highlighted in **bold**.

| Edit Pipelines | Method | $\text{CLIP}_d$ ($\downarrow$) | $\text{CLIP}_s$ ($\downarrow$) | FID ($\uparrow$) | $\text{F}_{1/8}$ ($\downarrow$) | $\text{F}_8$ ($\downarrow$) |
|---|---|---|---|---|---|---|
| | | | Face | | | |
| IP2P | No Protection | 0.1704 | 0.8730 | 76.2210 | 0.9882 | 0.9835 |
| | AdLift*-VU | **0.1586 (-0.0118)** | **0.7786 (-0.0944)** | **129.8172 (+53.5962)** | **0.8807 (-0.1075)** | **0.9279 (-0.0556)** |
| | AdLift*-VT | 0.1595 (-0.0109) | 0.8285 (-0.0445) | 93.4765 (+17.2555) | 0.9473 (-0.0409) | 0.9694 (-0.0141) |
| MagicBrush | No Protection | 0.1769 | 0.8635 | 78.1133 | 0.9784 | 0.9638 |
| | AdLift*-VU | **0.1475 (-0.0294)** | **0.7659 (-0.0976)** | **120.0269 (+41.9136)** | **0.9150 (-0.0634)** | **0.9145 (-0.0493)** |
| | AdLift*-VT | 0.1607 (-0.0162) | 0.8148 (-0.0487) | 100.0880 (+21.9747) | 0.9657 (-0.0127) | 0.9570 (-0.0068) |
| SDEdit | No Protection | 0.1708 | 0.7727 | 89.2094 | 0.9568 | 0.9722 |
| | AdLift*-VU | **0.1554 (-0.0154)** | 0.7397 (-0.033) | 104.9600 (+15.7506) | 0.9618 (0.0050) | 0.9575 (-0.0147) |
| | AdLift*-VT | 0.1751 (0.0043) | **0.7353 (-0.0374)** | **118.0674 (+28.8580)** | **0.8806 (-0.0762)** | **0.9443 (-0.0279)** |
| SDXL-IP2P | No Protection | 0.1140 | 0.7012 | 93.6249 | 0.9707 | 0.9672 |
| | AdLift*-VU | 0.1094 (-0.0046) | **0.6481 (-0.0531)** | **115.9670 (+22.3421)** | **0.9214 (-0.0493)** | 0.9289 (-0.0383) |
| | AdLift*-VT | **0.1081 (-0.0059)** | 0.6693 (-0.0319) | 112.9657 (+19.3408) | 0.9383 (-0.0324) | **0.9274 (-0.0398)** |
| | | | Fangzhou | | | |
| IP2P | No Protection | 0.1938 | 0.9053 | 51.7357 | 0.9905 | 0.9913 |
| | AdLift*-VU | 0.1791 (-0.0147) | **0.8037 (-0.1016)** | **102.6782 (+50.9425)** | **0.8337 (-0.1568)** | 0.8794 (-0.1119) |
| | AdLift*-VT | **0.1788 (-0.0150)** | 0.8148 (-0.0905) | 86.0002 (+34.2645) | 0.8862 (-0.1043) | **0.8676 (-0.1237)** |
| MagicBrush | No Protection | 0.1812 | 0.8566 | 73.9738 | 0.9032 | 0.8124 |
| | AdLift*-VU | 0.1605 (-0.0207) | **0.7608 (-0.0958)** | **144.9615 (+70.9877)** | 0.7101 (-0.1931) | 0.6113 (-0.2011) |
| | AdLift*-VT | **0.1581 (-0.0231)** | 0.7998 (-0.0568) | 124.9021 (+50.9283) | **0.6838 (-0.2194)** | **0.5643 (-0.2481)** |
| SDEdit | No Protection | 0.1220 | 0.7578 | 68.4325 | 0.9802 | 0.9621 |
| | AdLift*-VU | 0.1292 (0.0072) | 0.7300 (-0.0278) | 124.4493 (+56.0168) | 0.8783 (-0.1019) | 0.9191 (-0.0430) |
| | AdLift*-VT | **0.1194 (-0.0026)** | **0.6755 (-0.0823)** | **165.0488 (+96.6163)** | **0.6478 (-0.3324)** | **0.6729 (-0.2892)** |
| SDXL-IP2P | No Protection | 0.0738 | 0.7496 | 75.9508 | 0.9628 | 0.9113 |
| | AdLift*-VU | **0.0695 (-0.0043)** | 0.7321 (-0.0175) | **100.9508 (+25.0000)** | 0.9127 (-0.0501) | 0.9112 (-0.0001) |
| | AdLift*-VT | 0.0715 (-0.0023) | 0.7331 (-0.0165) | 93.3964 (+17.4456) | 0.9346 (-0.0282) | **0.8424 (-0.0689)** |

Moreover, stronger purification methods like Xue & Chen (2024) may further weaken adversarial perturbations, but AdLift remains compatible with purification-robust adversarial methods such as BlurGuard (Kim et al., 2026), allowing our framework to further improve robustness against purification.

*Table 13.* Robustness of `AdLift` against purification methods (training views). Bracketed values indicate changes relative to the unprotected baseline.

| Method | $\text{CLIP}_d$ ($\downarrow$) | $\text{CLIP}_s$ ($\downarrow$) | FID ($\uparrow$) | $\text{F}_{1/8}$ ($\downarrow$) | $\text{F}_8$ ($\downarrow$) |
|---|---|---|---|---|---|
| | | Face | | | |
| No Protection | 0.1685 | 0.8852 | 22.0270 | 0.9958 | 0.9947 |
| AdLift*-VU | 0.1554 (-0.0131) | 0.7620 (-0.1232) | 88.5865 (+66.5595) | 0.8954 (-0.1004) | 0.8742 (-0.1205) |
| AdLift*-VU-JPEG | 0.1548 (-0.0137) | 0.7851 (-0.1001) | 69.2879 (+47.2609) | 0.9375 (-0.0583) | 0.9359 (-0.0588) |
| AdLift*-VU-DiffPure | 0.1549 (-0.0136) | 0.8173 (-0.0679) | 51.5645 (+29.5375) | 0.9753 (-0.0205) | 0.9727 (-0.0220) |
| | | Fangzhou | | | |
| No Protection | 0.1937 | 0.9080 | 23.7996 | 0.9817 | 0.9795 |
| AdLift*-VU | 0.1781 (-0.0156) | 0.7900 (-0.1180) | 84.2180 (+60.4184) | 0.8030 (-0.1787) | 0.7749 (-0.2046) |
| AdLift*-VU-JPEG | 0.1791 (-0.0146) | 0.8218 (-0.0862) | 61.7341 (+37.9345) | 0.9197 (-0.0620) | 0.9332 (-0.0463) |
| AdLift*-VU-DiffPure | 0.1860 (-0.0077) | 0.8640 (-0.0440) | 35.8747 (+12.0751) | 0.9777 (-0.0040) | 0.9670 (-0.0125) |

*Table 14.* Robustness of `AdLift` against purification methods (novel views). Bracketed values indicate changes relative to the unprotected baseline.

| Method | $\text{CLIP}_d$ ($\downarrow$) | $\text{CLIP}_s$ ($\downarrow$) | FID ($\uparrow$) | $\text{F}_{1/8}$ ($\downarrow$) | $\text{F}_8$ ($\downarrow$) |
|---|---|---|---|---|---|
| | | Face | | | |
| No Protection | 0.1704 | 0.8730 | 76.2210 | 0.9882 | 0.9835 |
| AdLift*-VU | 0.1586 (-0.0118) | 0.7786 (-0.0944) | 129.8172 (+53.5962) | 0.8807 (-0.1075) | 0.9279 (-0.0556) |
| AdLift*-VU-JPEG | 0.1573 (-0.0131) | 0.7947 (-0.0783) | 116.7051 (+40.4841) | 0.9420 (-0.0462) | 0.9486 (-0.0349) |
| AdLift*-VU-DiffPure | 0.1573 (-0.0131) | 0.8162 (-0.0568) | 107.5986 (+31.3776) | 0.9639 (-0.0243) | 0.9432 (-0.0403) |
| | | Fangzhou | | | |
| No Protection | 0.1938 | 0.9053 | 51.7357 | 0.9905 | 0.9913 |
| AdLift*-VU | 0.1791 (-0.0147) | 0.8037 (-0.1016) | 102.6782 (+50.9425) | 0.8337 (-0.1568) | 0.8794 (-0.1119) |
| AdLift*-VU-JPEG | 0.1822 (-0.0116) | 0.8229 (-0.0824) | 85.8403 (+34.1046) | 0.9353 (-0.0552) | 0.9519 (-0.0394) |
| AdLift*-VU-DiffPure | 0.1883 (-0.0055) | 0.8618 (-0.0435) | 64.7646 (+13.0289) | 0.9756 (-0.0149) | 0.9687 (-0.0226) |

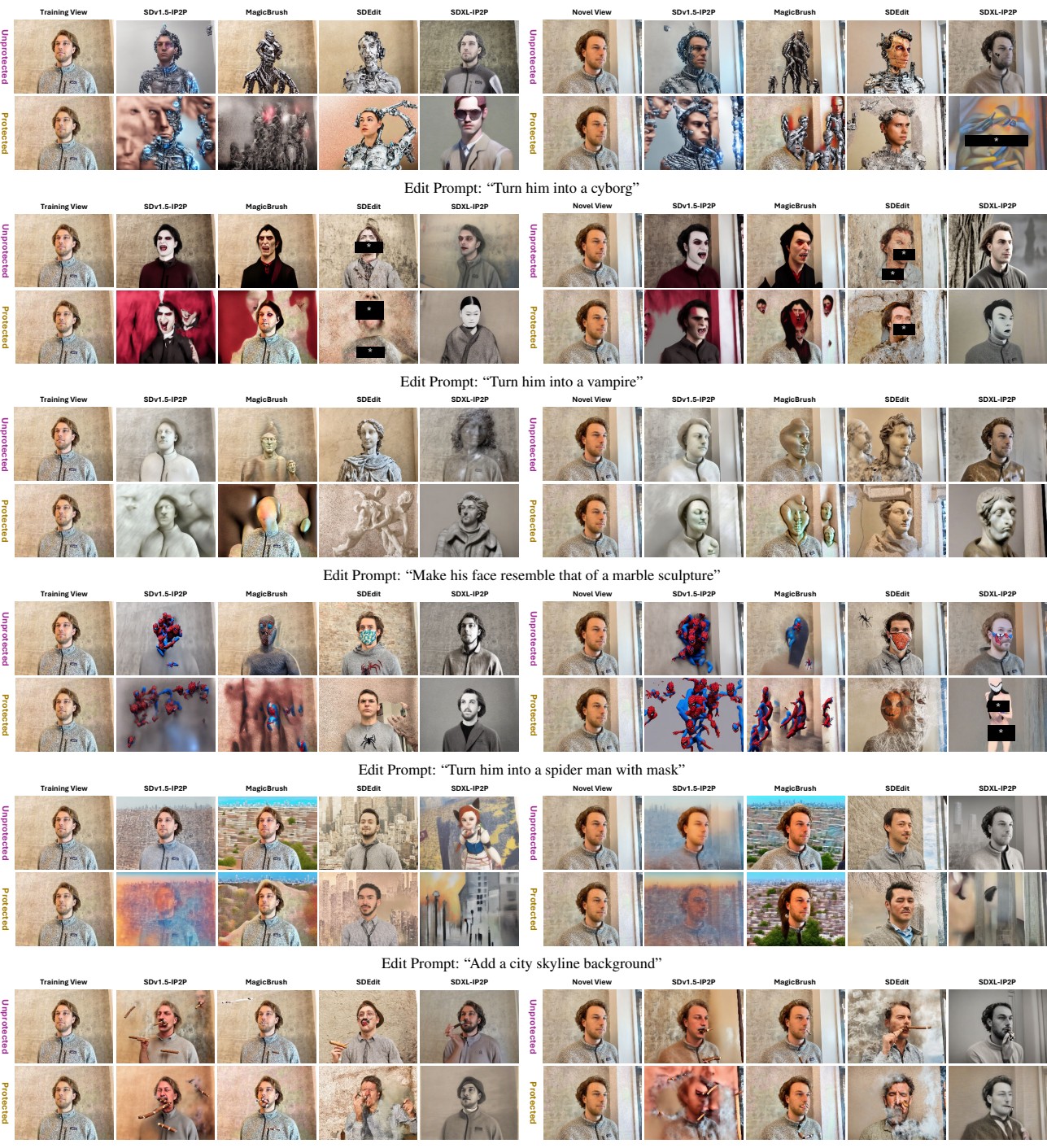

*Figure 31.* Transferability of AdLift. Warning: Following safety and ethical guidelines, harmful or sensitive content (e.g., sexual imagery, or violence) is masked in the figures.

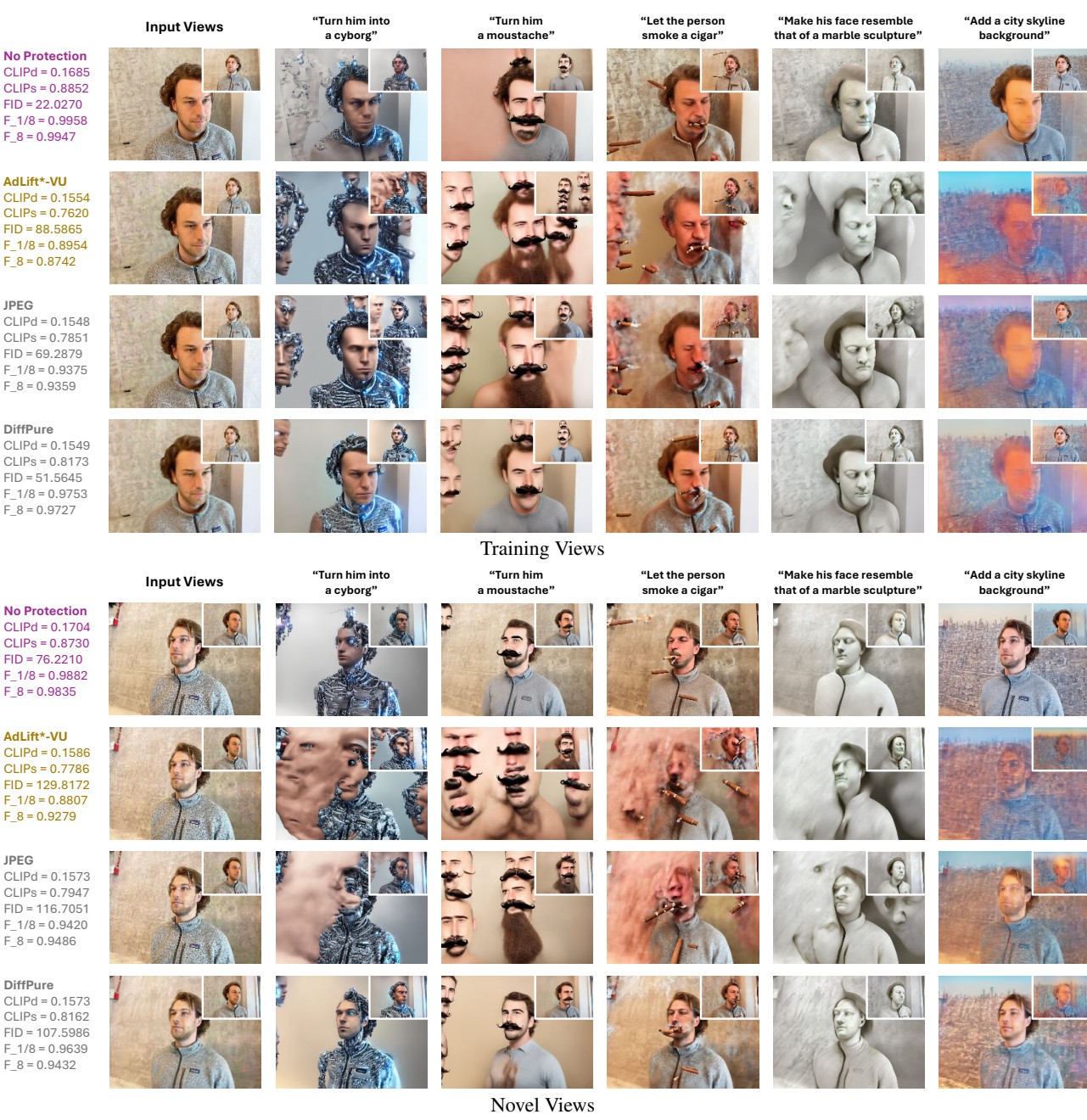

*Figure 32.* Robustness of `AdLift` against purification.

# G. More Experimental Details

## G.1. Evaluation Metrics

**Invisibility.** We assess the visual fidelity of rendered views against the ground truth using three standard metrics: Peak Signal-to-Noise Ratio (PSNR), Structural Similarity Index (SSIM) (Wang et al., 2004), and Learned Perceptual Image Patch Similarity (LPIPS) (Zhang et al., 2018). Specifically, higher SSIM and PSNR, and lower LPIPS indicate better fidelity to the original image.

**Protection Capability.** To measure protection effectiveness, we adopt two CLIP-based metrics: CLIP Text-Image Directional Similarity (denoted as $\text{CLIP}_d$) and CLIP Image Similarity ($\text{CLIP}_s$).

$\text{CLIP}_d$ evaluates how well the edit aligns with the input text instruction (Wang et al., 2024b; Chen et al., 2024b; Wu et al., 2024; Wang et al., 2025), and is computed as:

$$\text{CLIP}_d = \text{CosineSimilarity}(C_i(x^{\text{edit}}) - C_i(x), C_t(t^{\text{out}}) - C_t(t^{\text{in}})), \tag{10}$$

where $C_i$ and $C_t$ denote the image and text encoders of CLIP, respectively. For each edit instruction, we predefine the corresponding input description $t^{\text{in}}$ and target description $t^{\text{out}}$ (details in Appendix G.2).

$\text{CLIP}_s$ measures the similarity between edits applied to original and protected assets, defined as:

$$\text{CLIP}_s = \text{CosineSimilarity}\big(C_i(x_{\text{prot}}^{\text{edit}}), C_i(x_{\text{raw}}^{\text{edit}})\big). \tag{11}$$

In our setting, lower $\text{CLIP}_d$ or $\text{CLIP}_s$ values indicate stronger protection, since edits on protected assets deviate from instruction-following or from the corresponding edits on unprotected assets.

## G.2. Editing Instructions

Editing instructions are partially collected from prior 3DGS editing works (Wu et al., 2024; Wang et al., 2024b; Chen et al., 2024b; Wu et al., 2024; Wang et al., 2025) and further extended using GPT-5, followed by careful human review. Finally, 25 high-quality instructions are curated for each scene, as shown in Tables 15 to 17.

For both IP2P-based 2D and 3D editing, these instructions serve as the editing inputs. When computing $\text{CLIP}_d$, the corresponding source and target description are encoded into text embeddings within the CLIP space, which are then compared with image embeddings to evaluate semantic alignment, as shown in Equation (10).

*Table 15.* Editing instructions and source/target descriptions for two face scenes.

| Source Description | Target Description | Editing Instruction |
|---|---|---|
| a photo of a face of a man | a photo of a face of a Joker with green hair | turn him into a Joker with green hair |
| a photo of a face of a man | a photo of a face of an old man with wrinkles | turn him into an old man with wrinkles |
| a photo of a face of a man | a photo of a werewolf | Turn him into a werewolf |
| a photo of a face of a man | a photo of a cyborg | Turn him into a cyborg |
| a photo of a face of a man | a photo of a marble sculpture | Make his face resemble that of a marble sculpture |
| a photo of a face of a man | a photo of a face of a vampire | Turn him into a vampire |
| a photo of a face of a man | a photo of a face of a spider man with Mask | Turn him into a spider man with Mask |
| a photo of a face of a man | a photo of a face of a man with a moustache | turn him a moustache |
| a photo of a face of a man | a photo of a man wearing a pair of glasses | give him a pair of glasses |
| a photo of a face of a man | a photo of a face of a man with red hair | give him red hair |
| a photo of a face of a man | a photo of a face of a man with pink hair | Turn the person's hair pink |
| a photo of a face of a man | a photo of a face of a man with a tattoo | Let the person have a tattoo |
| a photo of a face of a man | a photo of a face of a man with elf-like ears | Add elf-like ears |
| a photo of a face of a man | a photo of a man wearing sunglasses | Let the person wear sunglasses |
| a photo of a face of a man | a photo of a face of a man smoking a cigar | Let the person smoke a cigar |
| a photo of a face of a man | a photo of a face of a man wearing a tiara | Place a tiara on the top of the head |
| a photo of a face of a man | a photo of a face of a man with a beach background | Change the background to a beach |
| a photo of a face of a man | a photo of a face of a man in a library background | Set the background in a library |
| a photo of a face of a man | a photo of a face of a man with a city skyline background | Add a city skyline background |
| a photo of a face of a man | a photo of a face of a man in a desert background | Change the background to a desert |
| a photo of a face of a man | a photo of a face of a man under a starry night sky | Set the background as a night sky with stars |
| a photo of a face of a man | a photo of a face of a man in a snowy forest | Set the background in a snowy forest |
| a photo of a face of a man | a photo of a face of a man in an office background | Put the man in an office background |
| a photo of a face of a man | a photo of a face of a man with a mountain background | Set the background to a mountain landscape |
| a photo of a face of a man | a photo of a face of a man standing under the moonlight | Let the person stand under the moon |

*Table 16.* Editing instructions and source/target descriptions for the bear scene.

| Source Description | Target Description | Editing Instruction |
|---|---|---|
| a photo of a bear statue in the forest | a photo of a grizzly bear in the forest | turn the bear into a grizzly bear |
| a photo of a bear statue in the forest | a photo of a polar bear in the forest | turn the bear into a polar bear |
| a photo of a bear statue in the forest | a photo of a panda in the forest | turn the bear into a panda |
| a photo of a bear statue in the forest | a photo of a golden bear in the forest | turn the bear into a golden bear |
| a photo of a bear statue in the forest | a photo of a robot bear in the forest | turn the bear into a robot bear |
| a photo of a bear statue in the forest | a photo of a skeleton bear in the forest | turn the bear into a skeleton bear |
| a photo of a bear statue in the forest | a photo of a bronze bear statue in the forest | turn the bear into a bronze statue |
| a photo of a bear statue in the forest | a photo of a glass bear in the forest | turn the bear into a glass sculpture |
| a photo of a bear statue in the forest | a photo of a wooden bear in the forest | turn the bear into a wooden bear |
| a photo of a bear statue in the forest | a photo of a giraffe in the forest | turn the bear into a giraffe |
| a photo of a bear statue in the forest | a photo of a fox in the forest | turn the bear into a fox |
| a photo of a bear statue in the forest | a photo of a raccoon in the forest | turn the bear into a raccoon |
| a photo of a bear statue in the forest | a photo of a bear statue in the snow | make the forest snowy |
| a photo of a bear statue in the forest | a photo of a bear statue in the forest at night | make it a night scene |
| a photo of a bear statue in the forest | a photo of a bear in the forest in the rain | make it raining |
| a photo of a bear statue in the forest | a photo of a bear statue in the forest during a storm | make it a stormy weather |
| a photo of a bear statue in the forest | a photo of a bear statue in the forest with a rainbow | add a rainbow in the background |
| a photo of a bear statue in the forest | a photo of a bear underwater | make it underwater |
| a photo of a bear statue in the forest | a photo of a bear in the desert | place the bear in a desert |
| a photo of a bear statue in the forest | a photo of a bear statue with fireflies in the forest | add fireflies around the bear |
| a photo of a bear statue in the forest | a photo of a mossy bear statue in the forest | add vines growing on the bear |
| a photo of a bear statue in the forest | a photo of a bear statue wearing a crown in the forest | put a crown on the bear |
| a photo of a bear statue in the forest | a photo of a chocolate bear in the forest | turn the bear into a chocolate sculpture |
| a photo of a bear statue in the forest | a photo of a crystal bear in the forest | turn the bear into a bear made of crystal |
| a photo of a bear statue in the forest | a photo of a fiery bear in the forest | turn the bear into a bear made of fire |

*Table 17.* Editing instructions and source/target descriptions for the stone-horse scene.

| Source Description | Target Description | Editing Instruction |
|---|---|---|
| a photo of a stone horse statue in front of the museum | a photo of a grizzly bear in front of the museum | turn the stone horse into a grizzly bear |
| a photo of a stone horse statue in front of the museum | a photo of a giraffe in front of the museum | turn the stone horse into a giraffe |
| a photo of a stone horse statue in front of the museum | a photo of a red panda in front of the museum | turn the stone horse into a red panda |
| a photo of a stone horse statue in front of the museum | a photo of a robot horse in front of the museum | turn the stone horse into a robot horse |
| a photo of a stone horse statue in front of the museum | a photo of a polar bear in front of the museum | turn the stone horse into a polar bear |
| a photo of a stone horse statue in front of the museum | a photo of a golden horse in front of the museum | turn the stone horse into a golden horse |
| a photo of a stone horse statue in front of the museum | a photo of a red horse in front of the museum | turn the stone horse into a red horse |
| a photo of a stone horse statue in front of the museum | a photo of a horse under the water in front of the museum | make it under the water |
| a photo of a stone horse statue in front of the museum | a photo of a horse in the snow in front of the museum | make it snowy |
| a photo of a stone horse statue in front of the museum | a photo of a horse at night in front of the museum | make it at night |
| a photo of a stone horse statue in front of the museum | a photo of a horse in the storm in front of the museum | make it in the storm |
| a photo of a stone horse statue in front of the museum | a photo of a zebra in front of the museum | turn the stone horse into a zebra |
| a photo of a stone horse statue in front of the museum | a photo of a unicorn in front of the museum | turn the stone horse into a unicorn |
| a photo of a stone horse statue in front of the museum | a photo of a pegasus in front of the museum | turn the stone horse into a pegasus |
| a photo of a stone horse statue in front of the museum | a photo of a knight riding the horse in front of the museum | add a knight riding the horse |
| a photo of a stone horse statue in front of the museum | a photo of a mossy horse statue in front of the museum | cover the horse with moss |
| a photo of a stone horse statue in front of the museum | a photo of an ice horse in front of the museum | turn the horse into ice sculpture |
| a photo of a stone horse statue in front of the museum | a photo of a graffiti-covered horse statue in front of the museum | paint the horse statue with graffiti |
| a photo of a stone horse statue in front of the museum | a photo of a horse in the desert in front of the museum | place the horse in the desert |
| a photo of a stone horse statue in front of the museum | a photo of a horse skeleton in front of the museum | turn the horse into a skeleton |
| a photo of a stone horse statue in front of the museum | a photo of a horse statue decorated with fairy lights in front of the museum | add fairy lights on the horse |
| a photo of a stone horse statue in front of the museum | a photo of a crowned horse in front of the museum | put a crown on the horse |
| a photo of a stone horse statue in front of the museum | a photo of a winged horse in front of the museum | add wings to the horse |
| a photo of a stone horse statue in front of the museum | a photo of a horse statue surrounded by butterflies in front of the museum | add butterflies around the horse |
| a photo of a stone horse statue in front of the museum | a photo of a glass horse in front of the museum | make the horse transparent like glass |

## G.3. More Details on Attacking Segment-Anything

In Equation (9), $\mathcal{L}_{\text{BCE}}$ and $\mathcal{L}_{\text{Dice}}$ are used in the local editing variant (`AdLift`-ST), where segmentation models (e.g., SAM (Kirillov et al., 2023)) are used to localize editable regions. We therefore adopt these complementary losses to deliberately disrupt mask prediction and prevent correct region localization. Specifically:

- $\mathcal{L}_{\text{BCE}}$ (Binary Cross-Entropy): encourages accurate pixel-wise alignment between the predicted and target masks.

- $\mathcal{L}_{\text{Dice}}$ (Dice Loss): encourages region-level alignment by maximizing the overlap between predicted and target masks.

# H. Discussions

**Definition of suppression.** The protection goal of `AdLift` is to suppress the effectiveness of instruction-driven editing (ensure that instruction-driven manipulation on the protected 3DGS asset becomes unreliable or unusable). Following the definition in prior 2D editing-guard works (Salman et al., 2023; Chen et al., 2024c; Van Le et al., 2023; Liu et al., 2024; Wang et al., 2024a; Xue et al., 2024; Choi et al., 2025), an edit is considered suppressed if the edited output either: *fails to follow the given editing instruction* (i.e., semantic inconsistency), or *becomes implausible or identity-inconsistent compared to editing the unprotected asset* (i.e., degraded realism or broken structure).

**Difference compared to existing 2D and 3DGS IP protection methods.** Our method is inspired by adversarial optimization (e.g., PGD (Madry et al., 2017)) but introduces a different problem formulation and optimization structure tailored to 3DGS. Specifically, safeguarding 3DGS assets introduces unique challenges that do not arise in standard 2D adversarial settings:

- *View inconsistency of 2D perturbations.* Adversarial perturbations optimized independently in 2D are inherently view-specific. When directly lifted via 3DGS reconstruction, they introduce cross-view conflicts, leading to underfitting and poor generalization to novel views (Figure 2 and Figure 10 in our manuscript).

- *Limitations of soft constraints in 3DGS.* Existing protection methods for 3DGS assets (e.g., 3DGS watermarking (Huang et al., 2024; Chen et al., 2025; Zhao et al., 2025)) rely on soft regularization in Gaussian parameter space, which is insufficient for adversarial objectives. As shown in our experiments (Figures 3, 9 and 11 to 14, Tables 3 and 4 to 7), they fail to achieve a satisfactory trade-off between invisibility and protection strength.

- *Absence of a unified perturbation budget.* Unlike pixels in 2D images, 3DGS consists of heterogeneous Gaussian attributes (position, scale, opacity, SH coefficients) (Kerbl et al., 2023; Zhang et al., 2026), making it non-trivial to define a unified perturbation budget and directly apply PGD-style projection (Goodfellow et al., 2014; Madry et al., 2017).

To address these challenges, we reformulate 3DGS protection as a *cross-space constrained optimization problem* (i.e., Equation (3)). Our key contributions and distinctions from existing gradient ascent PGD are:

- *Different problem formulation.* We optimize perturbations in the 3D Gaussian space, while enforcing strict constraints in the multi-view 2D rendering space. This creates a fundamental mismatch between the optimization space and the constraint space, which, to the best of our knowledge, has not been explicitly addressed in prior 2D editing guard and 3DGS copyright protection works.

- *Different optimization structure.* To solve this cross-space problem, we propose a novel two-stage alternating optimization Lifted PGD (L-PGD), which consists of:
  - *Gradient truncation in rendering space,* which enforces hard perceptual bounds in the intermediate rendering space for strict invisibility constraints;
  - *Image-to-Gaussian fitting,* which propagates these constrained perturbations back into the 3D Gaussian parameters.
  This *lift-and-fit* mechanism is different from standard PGD: rather than projecting in a fixed space, we iteratively bridge two different spaces (2D and 3D) to ensure both perceptual control and cross-view consistency.

Importantly, this design is necessary for enabling adversarial protection in 3DGS:

- Direct 2D PGD fails due to view inconsistency,

- Direct 3D optimization fails due to lack of perceptual control,

- Our cross-space formulation and L-PGD can simultaneously achieve view-consistency, strict invisibility, and protection strength to editing models.

**The choice of perturbation budget** $\eta$**.** We provide an ablation study in Figure 26, and based on the results, we recommend $\eta = 16$ as the default budget in practice, as it achieves strong protection while maintaining reasonable visual quality. Notably, this choice is also aligned with commonly adopted settings in adversarial attack and editing protection literature like (Xue et al., 2024; Choi et al., 2025). Besides, for applications where imperceptibility is critical (e.g., portrait or facial assets), a smaller budget such as $\eta = 8$ offers a more conservative choice. On the other hand, for scenarios prioritizing maximum protection where some visual artifacts are acceptable, $\eta = 24$ can further enhance robustness.

**Convergence guarantees and accumulated errors.** The image-to-Gaussian fitting step may introduce approximation error, as it does not correspond to an exact reconstruction. However, unlike standard 3DGS reconstruction, which relies on fixed multi-view ground-truth supervision, adversarial perturbations do not have a known optimal target in either the 3D Gaussian space or the rendering space. As such, to obtain effective adversarial perturbations, at each iteration we construct a locally

improved adversarial target in the rendering space (i.e., a point with higher adversarial loss for the current view), and update the Gaussian parameters to align with this target. Notably, this process is dynamic. The adversarial objective is re-computed at every iteration through rendering and gradient updates, and the optimization is performed across training views. As a result, although each fitting step introduces approximation error, the Gaussian-represented perturbations are continuously corrected as the 3D representation is re-aligned with updated adversarial objectives, leading to progressively stronger attack effectiveness. Empirically, this is supported by the training dynamics in Figure 10, where the adversarial loss consistently improves over iterations on both training and novel views, without degradation in later stages. This indicates that the adversarial perturbation is progressively strengthened. Besides, while our current work focuses on empirical validation, we believe that establishing theoretical guarantees for such cross-space optimization is an important direction for future research.

## I. Limitations and Future Directions

**Further balance invisibility against protection effectiveness.** `AdLift` exhibits a better balance compared to broader baselines, consistently achieving stronger editing resistance while maintaining similar or better perceptual quality. This also indicates that the learned perturbations are meaningful rather than trivial noise. However, subtle visual artifacts may still appear. As such, further reducing perceptual impact while retaining protection strength remains an important direction.

**Computation overhead.** Lifting PGD into the 3DGS space introduces additional computational overhead. The increased cost is mainly due to the need to back-propagate gradients through the editing model on multiple rendered views in order to achieve view-generalizable protection. The computation scales with both the editing model complexity and the number of supervised viewpoints. This reflects an inherent challenge of adversarial learning for 3DGS and highlights opportunities for future efficiency improvements, such as using lightweight surrogate editing models or adaptive view-sampling strategies.

**Robustness to purification.** Our findings in Appendix F.2 show that `AdLift` exhibits similar behavior to 2D adversarial protection: they remain effective under JPEG compression-based purification and generalize across unseen views. While diffusion-based purification reduces protection strength, `AdLift` still outperforms unprotected 3DGS assets. We consider improving purification robustness, potentially by lifting purification-resilient 2D adversarial strategies into the 3D domain, as an important direction for future work.

