# OpenReview forum: "AdLift: Lifting Adversarial Perturbations to Safeguard 3D Gaussian Splatting Assets Against Instruction-Driven Editing"
_ICML.cc/2026/Conference — ICML 2026 spotlight_

### Official Review · Reviewer_82ur · 2026-03-05

**Soundness:** 3
**Presentation:** 2
**Significance:** 3
**Originality:** 3
**Overall Recommendation:** 4
**Confidence:** 5

**Summary:**

This paper proposes AdLift, a method to protect 3D Gaussian Splatting (3DGS) assets from instruction-driven editing models. The approach generates adversarial perturbations in the 2D rendered image space and then lifts them back into the 3D Gaussian representation using a Lifted PGD optimization scheme. Experiments show the method reduces the effectiveness of both 2D and 3D editing pipelines while maintaining visual fidelity.

**Compliance With Llm Reviewing Policy:**

Affirmed.

**Final Justification:**

The rebuttal has adequately addressed my concerns.

**Key Questions For Authors:**

Please refer to **W1** and **W2**.

**Limitations:**

yes

**Strengths And Weaknesses:**

## Strengths

1. The paper is clearly written and easy to follow. The experimental section is particularly well organized.

2. The paper proposes a highly customized optimization method, **LiftedPGD**, and adopts an alternating optimization strategy to achieve a trade-off between the **visual imperceptibility of perturbations** and **attack effectiveness**.

---

## Weaknesses

1. The core contribution of the paper mainly lies in designing a customized **LiftedPGD** based on the specific characteristics of **3D Gaussian Splatting assets**, and obtaining the final perturbations through an alternating optimization strategy. This essentially introduces a new optimization procedure rather than proposing a fundamentally new attack paradigm.

2. The paper does not discuss any **convergence guarantees** for the proposed optimization method. In particular, in the **Image-to-Gaussian fitting** stage, the update of the Gaussian parameters $\mathcal{G}^k$ is intended to make the rendered projection approximate  $R(\hat{\mathcal{G}}^{k+1}, v^t).$ However, this approximation does not correspond to an exact reconstruction, and some reconstruction error is inevitable. Such approximation errors may accumulate and potentially lead to significant degradation in the final attack effectiveness.

3. There are several **inconsistencies in notation** throughout the paper. For example, at line 192 the notation $G^{src}$ and $G^{edit}$ are used, while at line 238 the paper instead uses $G^{prot}$ and $G^{raw}$. Moreover, below Equation (4) the notation changes back to $G^{src}$. The authors should carefully check the notation and ensure consistency throughout the manuscript.

4. The losses $L_{BCE}$ and $L_{Dice}$ in Equation (9) are not explained in sufficient detail in the paper, which makes this part difficult to understand.

---

> ### Author Rebuttal · Authors · 2026-03-31
>
> Dear Reviewer `82ur`,
>
> Thank you for the insightful review. Please see our responses below.
>
>
> > **W1: The contribution**
>
>
> **R1:** We thank the reviewer for this point. Our method is inspired by adversarial optimization (e.g., PGD) but introduces a different problem formulation and optimization structure for 3DGS.
>
>
> - **Existing works** either:
>     - optimize perturbations independently in 2D, leading to severe cross-view inconsistency and underfitting (Fig. 2, Fig. 10 in our manuscript);
>     - impose soft constraints directly in the Gaussian parameter space, which cannot effectively balance invisibility and protection strength (Figs 3, 9, 11-14, Tabs. 3–7);
>
>     These limitations are inherent: 2D methods lack 3D consistency, while 3D parameter-space soft constraints cannot provide precise perceptual control. Beside, 3DGS parameters consist of heterogeneous attributes, it is non-trivial to impose a unified perturbation budget analogous to 2D pixel space, hence standard PGD cannot be directly applied.
>
> - **Our method:** To resolve the above limitations, we propose L-PGD.
>     - L-PGD reformulates 3DGS protection as a *cross-space constrained optimization problem*, where the optimization variables reside in the 3D Gaussian space, while the constraints are defined in the multi-view 2D rendering space.
>     - To solve this cross-space problem, we propose a novel two-stage alternating optimization:
>         - enforcing strict constraints in the rendering space via gradient truncation and projected updates, and
>         - lifting these constrained perturbations back to the 3D Gaussian space via image-to-Gaussian fitting.
>
>     This design bridges the gap between 2D perceptual constraints and 3D representations, enabling view-consistent and generalizable protection that cannot be achieved by existing 2D-based or 3D-parameter-based approaches.
>
> We will further emphasize this problem formulation and its distinction from standard PGD-based methods in the revision.
>
>
> > **W2: Convergence guarantees and accumulate errors**
>
>
> **R2:** We thank the reviewer for this insightful review. We agree that the image-to-Gaussian fitting step introduces approximation error, as it does not correspond to an exact reconstruction. However, unlike standard 3DGS reconstruction, which relies on fixed multi-view ground-truth supervision, *adversarial perturbations do not have a known optimal target* in either the 3D Gaussian space or the rendering space. As such, to obtain effective adversarial perturbations, at each iteration we construct a *locally improved adversarial target* in the rendering space (i.e., a point with higher adversarial loss for the current view), and update the Gaussian parameters to align with this target. Notably, this process is dynamic. The adversarial objective is re-computed at every iteration through rendering and gradient updates, and the optimization is performed across training views. As a result, although each fitting step introduces approximation error, the Gaussian-represented perturbations are continuously corrected as the 3D representation is re-aligned with updated adversarial objectives, leading to progressively stronger attack effectiveness.
>
> **Empirically**, this is supported by the training dynamics in Fig. 10, where the adversarial loss consistently improves over iterations on both training and novel views, without degradation in later stages. This indicates that the adversarial perturbation is progressively strengthened.
>
> While our current work focuses on empirical validation, we believe that establishing theoretical guarantees for such cross-space optimization is an important direction for future research, and we will include additional discussion in the revision.
>
>
> > **W3: Notation inconsistencies**
>
> **R3:** We thank the reviewer for pointing this out. We carefully check the entire manuscript and identify all notation inconsistencies. In the revised version, we uniformly use $G^{\text{raw}}$ to denote the original unprotected asset and $G^{\text{prot}}$ to denote its protected version throughout the paper, ensuring full consistency across all sections.
>
>
> > **W4: Explaination of losses $L_{\text{BCE}}$ and $L_{\text{Dice}}$**
>
> **R4:** We are sorry for the confusion. $L_{\text{BCE}}$ and $L_{\text{Dice}}$ are used in the local editing variant (AdLift-ST), where segmentation models (e.g., SAM [D1]) are used to localize editable regions. We therefore adopt these complementary losses to deliberately disrupt mask prediction and prevent correct region localization. Specifically:
> - $L_{\text{BCE}}$ (Binary Cross-Entropy): encourages accurate pixel-wise alignment between the predicted and target masks.
> - $L_{\text{Dice}}$ (Dice Loss): encourages region-level alignment by maximizing the overlap between predicted and target masks.
>
> We will include definitions and explanations of both losses in the revision.
>
>
> ---
>
> [D1] Segment Anything. ICCV 2023

---

> > ### Author Rebuttal · Reviewer_82ur · 2026-04-02
> >
> > We thank the authors for their response. However, we have one minor follow-up question:
> >
> > Have the authors conducted transferability evaluations or direct white-box optimization on recent DiT-based (Diffusion Transformer) image editing models?
> > Specifically, we wonder if the method has been evaluated on recent DiT-based editors (e.g., FLUX.1, SD3) under black-box or white-box settings. Such results would better demonstrate its generalizability across modern architectures.

---

> > > ### Author Response · Authors · 2026-04-07
> > >
> > > Dear Reviewer `82ur`,
> > >
> > > Thank you for acknowledging that your previous concerns have been addressed, and for the constructive follow-up.
> > >
> > > > **Follow-up Q1: Have the authors conducted transferability evaluations or direct white-box optimization on recent DiT-based (Diffusion Transformer) image editing models? Specifically, we wonder if the method has been evaluated on recent DiT-based editors (e.g., FLUX.1, SD3) under black-box or white-box settings. Such results would better demonstrate its generalizability across modern architectures.**
> > >
> > > **Follow-up R1**: Thank you for this constructive question. We conducted additional experiments using **SD3-UltraEdit** [D2-3], an instruction-based image editing model that follows the InstructPix2Pix-style paradigm with higher-quality and more diverse supervision, and is built on Stable Diffusion 3 with the MMDiT architecture for enhanced multimodal modeling. As suggested by the reviewer, in this setting, we adopt SD3-UltraEdit as the surrogate model for L-PGD optimization.
> > >
> > > The quantitative results are reported in **Tab. D1**, and qualitative results are available at the anonymous Github repo: https://anonymous.4open.science/r/ICML2026_ID11967_rebuttal-FB8B/SD3-UltraEdit.png.
> > >
> > > From the results, AdLift demonstrates consistently strong protection performance on the SD3-UltraEdit. Specifically, across all five editing quality metrics, the protected assets show consistent improvements in protection effectiveness compared to the unprotected baseline. Moreover, the visual results indicate that edited outputs from protected assets suffer from noticeable quality degradation or fail to preserve identity, further validating the effectiveness of AdLift under DiT-based editing.
> > >
> > > These results suggest that our L-PGD framework can be effectively applied to modern DiT-based editing models, demonstrating strong compatibility across different model architectures. We will include this discussion on the generalizability of AdLift across modern architectures in the revised version.
> > >
> > > **Table D1:** White-box results on SD3-UltraEdit (Face, budget=16/255). Bracketed values indicate changes relative to unprotected baseline.
> > >
> > > |View|Method|SSIM ($\uparrow$)|PSNR ($\uparrow$)|LPIPS ($\downarrow$)|CLIP_d($\downarrow$)|CLIP_s($\downarrow$)|FID ($\uparrow$)|F_1/8 ($\downarrow$)|F_8 ($\downarrow$)|
> > > |:-:|:-:|:-:|:-:|:-:|:-:|:-:|:-:|:-:|:-:|
> > > |Train|Unprotected|0.9381|32.6436|0.0668|0.1687|0.8576|20.3739|0.9958|0.9962|
> > > ||**AdLift-VU (DiT)**|0.6827|27.2159|0.4539|0.1505 (**-0.0182**)|0.7076 (**-0.1500**)|59.6826 (**+39.3087**)|0.9450 (**-0.0508**)|0.9195 (**-0.0767**)|
> > > |Novel|Unprotected|0.8590|26.0391|0.1292|0.1679|0.8306|78.3555|0.9060|0.9788|
> > > ||**AdLift-VU (DiT)**|0.6199|23.5639|0.4845|0.1552 (**-0.0127**)|0.7020 (**-0.1286**)|119.6532 (**+41.2977**)|0.9011 (**-0.0049**)|0.8451 (**-0.1337**)|
> > >
> > > ---
> > >
> > > [D2] https://huggingface.co/BleachNick/SD3_UltraEdit_w_mask
> > > [D3] UltraEdit: Instruction-based Fine-Grained Image Editing at Scale. NeurIPS 2024

---

### Official Review · Reviewer_6Qbd · 2026-03-09

**Soundness:** 4
**Presentation:** 4
**Significance:** 3
**Originality:** 4
**Overall Recommendation:** 5
**Confidence:** 3

**Summary:**

This paper investigates the problem of actively safeguarding 3D Gaussian Splatting (3DGS) assets against instruction-driven editing. With the rapid development of diffusion-model-enabled instruction-guided 2D and 3D editing techniques, 3DGS assets can be manipulated through arbitrary textual instructions, thereby exposing them to risks of unauthorized modification and malicious tampering. Existing approaches primarily focus on passive intellectual property protection strategies, such as watermarking and steganography, which are insufficient to defend against diffusion-based instruction-driven editing attacks.

To address this issue, the authors propose an active safeguarding method termed AdLift. The method first generates strictly bounded adversarial perturbations in the 2D rendering space. Through a tailored Lifted PGD (L-PGD) optimization framework, the constrained 2D perturbations are progressively mapped and updated into the 3D Gaussian representation by alternating between gradient truncation and image-to-Gaussian fitting. Specifically, gradient truncation is applied to strictly constrain perturbations at the rendered image level, and the resulting perturbations are then propagated to the Gaussian parameters via an image-to-Gaussian fitting operation. By iteratively performing these two steps, the method constructs a dedicated set of safeguard Gaussians in the 3D space, thereby suppressing instruction-driven editing on 3DGS assets.

Experimental results demonstrate that AdLift effectively mitigates instruction-driven editing across multiple scenes and diverse 2D/3D editing tasks, while maintaining strong visual imperceptibility.

**Compliance With Llm Reviewing Policy:**

Affirmed.

**Final Justification:**

The rebuttal has adequately addressed my concerns. This paper is worthy of publication at ICML.

**Key Questions For Authors:**

see Other Strengths And Weaknesses.

**Limitations:**

yes

**Strengths And Weaknesses:**

Strengths：
1. The paper studies a protection method for the risks of unauthorized editing and malicious tampering arising from the fact that 3D Gaussian Splatting (3DGS) assets can be modified by arbitrary text instructions. This problem is of practical relevance.
2. The authors impose strict constraints on the perturbations in the 2D rendering space and use Lifted PGD to map them into the 3D Gaussian parameter space, avoiding the need to directly impose complex constraints on Gaussian parameters. The overall optimization procedure is clear, and the technical pipeline is well defined.
3. The paper conducts experiments on multiple scenes and different 2D/3D editing methods, and also includes some transferability tests. The experimental results generally support the effectiveness of the proposed method.

Weaknesses：Several minor issues
1. The paper aims to suppress instruction-driven editing, but the current evaluation mainly relies on similarity metrics such as CLIP. It is still unclear how suppression is defined, that is, under what conditions an edit can be regarded as effectively suppressed.
2.  As shown in Figure 28, the perturbation budget (\eta) has a significant impact on the protection effect and reveals a clear trade-off between imperceptibility and protection strength. Although the paper shows the trends under different \eta values, it does not clearly specify a recommended budget range or explain how \eta should be chosen in practice.

---

> ### Author Rebuttal · Authors · 2026-03-31
>
> Dear Reviewer `6Qbd`,
>
> Thank you for your helpful comments. We address your concerns as follows.
>
> > **W1: The definition of "suppression" in editing guard**
>
> **R1:** Thank you for the question. We clarify the definition of suppression and the evaluation protocol and metrics.
>
> - **Definition of suppression.** We clarify that the protection goal of AdLift is to *suppress the effectiveness of instruction-driven editing* (ensure that instruction-driven manipulation on the protected 3DGS asset becomes unreliable or unusable). Following the definition in prior 2D editing-guard works [C1-C7], an edit is considered suppressed if the edited output either:
>     - **fails to follow the given editing instruction** (i.e., semantic inconsistency), or
>     - **becomes implausible or identity-inconsistent** compared to editing the unprotected asset (i.e., degraded realism or broken structure).
>
> - **Evaluation protocol.** Following prior works, we adopt a *relative evaluation protocol* by comparing editing results on *protected vs. unprotected* assets. Since suppression is inherently a *soft notion* (i.e., non-binary), we do not define a hard threshold. Instead, stronger suppression corresponds to larger divergence between the two. To comprehensively quantify suppression from the two perspectives, we adopt the following metrics [C8-C10]:
>
>     - **CLIP$_d$**: measures how well the edit follows the instruction (lower indicates stronger suppression).
>     - **CLIP$_s$**: measures similarity between edits on protected and unprotected assets (lower indicates stronger deviation).
>     - **FID**: evaluates distributional realism; higher values indicate degraded generation quality and thus stronger resistance to editing.
>     - **PRD** (including **F$_{1/8}$** and **F$_8$**): capture precision–recall trade-offs of generated distributions; lower values indicate stronger deviation from normal edited outputs.
>
> We will clarify this definition and evaluation protocol explicitly in the revised manuscript.
>
>
> > **W2: The choice of perturbation budget $\eta$**
>
>
> **R2:** Thank you for pointing this out. We provide an ablation study in Fig. 28, and based on the results, we recommend **$\eta$=16** as the **default budget** in practice, as it achieves strong protection while maintaining reasonable visual quality. Notably, this choice is also aligned with commonly adopted settings in adversarial attack and editing protection literature like [C5-C7]. Besides, for applications where imperceptibility is critical (e.g., portrait or facial assets), a smaller budget such as $\eta$=8 offers a more conservative choice. On the other hand, for scenarios prioritizing maximum protection where some visual artifacts are acceptable, $\eta$=24 can further enhance robustness.
>
> We will include this guideline and literature context in the revision.
>
>
> ---
>
> [C1] Raising the Cost of Malicious AI-Powered Image Editing. ICML 2023 \
> [C2] EditShield: Protecting Unauthorized Image Editing by Instruction-Guided Diffusion Models. ECCV 2024 \
> [C3] Anti-DreamBooth: Protecting Users from Personalized Text-to-image Synthesis. ICCV 2023 \
> [C4] MetaCloak: Preventing Unauthorized Subject-driven Text-to-image Diffusion-based Synthesis via Meta-learning. CVPR 2024 \
> [C5] SimAC: A Simple Anti-Customization Method for Protecting Face Privacy against Text-to-Image Synthesis of Diffusion Models. CVPR 2024 \
> [C6] Toward effective protection against diffusion-based mimicry through score distillation. ICLR 2024 \
> [C7] DiffusionGuard: A Robust Defense Against Malicious Diffusion-based Image Editing. ICLR 2025 \
> [C8] Instructpix2pix: Learning to follow image editing instructions. CVPR 2023 \
> [C9] GANs Trained by a Two Time-Scale Update Rule Converge to a Local Nash Equilibrium. NIPS 2017 \
> [C10] Assessing Generative Models via Precision and Recall. NeurIPS 2018

---

> > ### Author Rebuttal · Reviewer_6Qbd · 2026-04-01
> >
> > Thank the authors for the detailed response. The response adequately addresses my previous concerns. This paper is worthy of publication at ICML.

---

> > > ### Author Response · Authors · 2026-04-07
> > >
> > > Dear Reviewer `6Qbd`,
> > >
> > > Thank you for your acknowledgement and constructive feedback. We sincerely appreciate your recognition of our work and are glad that our clarifications have addressed your concerns.
> > >
> > > We will further improve the manuscript based on your suggestions in the revision.
> > >
> > > We sincerely thank you again for your time and valuable feedback.
> > >
> > > Best regards,
> > > Submission11967 Authors

---

### Official Review · Reviewer_FdAC · 2026-03-13

**Soundness:** 4
**Presentation:** 4
**Significance:** 4
**Originality:** 4
**Overall Recommendation:** 5
**Confidence:** 4

**Summary:**

### summary

This paper proposes AdLift, the first 3D adversarial defense framework designed to protect emerging 3D Gaussian Splatting (3DGS) assets against unauthorized, instruction-driven editing and manipulation. To overcome the inherent trade-off between cross-view generalizability and perturbation stealthiness when directly applying 2D adversarial perturbations to 3D spaces, the authors design a "Lifted Projected Gradient Descent (Lifted PGD)" strategy. By introducing gradient truncation at the rendered image level and alternating it with an Image-to-Gaussian fitting process, this strategy successfully "lifts" strictly bounded 2D defensive perturbations and implicitly embeds them into 3D Gaussian parameters. This method not only effectively balances visual quality with defense robustness but also demonstrates strong consistency and resilience against various advanced diffusion-based editing models across both training and novel views. Overall, it fills a crucial technical gap in the security protection of 3D AIGC assets.

**Compliance With Llm Reviewing Policy:**

Affirmed.

**Key Questions For Authors:**

In conclusion, I appreciate the authors for presenting this excellent piece of work. A rebuttal addressing the aforementioned weaknesses will help clarify the paper's true technical contributions. Based on my preliminary assessment, I would strongly recommend this paper as an Oral or Spotlight candidate. I gave it a initial score of 5 (instead of 6) because of the technical context of the work (ethical concerns or adversarial defenses), and it is not a general contribution to the machine learning community.

**Limitations:**

yes

**Strengths And Weaknesses:**

### Strengths

Overall, this paper explores a highly compelling topic. The authors have done a solid job in both presentation and experimental validation (especially the extensive qualitative sample analyses). Therefore, my initial rating is highly positive.

Originality -- Yes. As claimed by the authors, this is the first proactive protection framework tailored for 3DGS assets, holding pioneering significance in the field of 3D AIGC security.

Significance -- Strong. AdLift provides a highly inspiring solution and a solid baseline, which will very likely spearhead further follow-up research on adversarial defenses for 3D representations.

Presentation -- Excellent. The overall narrative is logically rigorous, the motivation is stated straightforwardly, and the writing is highly fluent, significantly lowering the reading barrier.

### Weaknesses

Despite the novel application scenario, AdLift's core algorithmic contributions appear somewhat limited. The proposed "customized Lifted PGD" seems to be merely an incremental migration of the standard PGD algorithm. Similar optimization strategies are quite common in the adversarial attack/defense literature for multi-view reconstruction.

Therefore, it is necessary for the authors to further clarify their technical contributions. The authors need to explicitly elaborate on the fundamental differences between their method and a generic "gradient ascent PGD + gradient truncation" strategy, highlighting the core innovations specifically tailored to the 3DGS architecture.

---

> ### Author Rebuttal · Authors · 2026-03-31
>
> Dear Reviewer `FdAC`,
>
> We appreciate the helpful review and provide detailed responses below.
>
>
> > **W1: Algorithmic contributions**
>
>
> **R1:** Thank you for the question. We here clarify that our method is inspired by adversarial optimization (e.g., PGD) but introduces a different problem formulation and optimization structure tailored to 3DGS. Specifically, safeguarding 3DGS assets introduces unique challenges that do not arise in standard 2D adversarial settings:
> - **View inconsistency of 2D perturbations.** Adversarial perturbations optimized independently in 2D are inherently view-specific. When directly lifted via 3DGS reconstruction, they introduce cross-view conflicts, leading to underfitting and poor generalization to novel views (Fig. 2, Fig. 10 in our manuscript).
> - **Limitations of soft constraints in 3DGS.** Existing protection methods for 3DGS assets (e.g., 3DGS watermarking) rely on soft regularization in Gausian parameter space, which is insufficient for adversarial objectives. As shown in our experiments (Figs 3, 9, 11-14, Tabs. 3–7), they fail to achieve a satisfactory trade-off between invisibility and protection strength.
> - **Absence of a unified perturbation budget.** Unlike pixels in 2D image, 3DGS consists of heterogeneous Gaussian attributes (position, scale, opacity, SH coefficients), making it non-trivial to define a unified perturbation budget and directly apply PGD-style projection.
>
>
> To address these challenges, we *reformulate 3DGS protection as a cross-space constrained optimization problem*. Our key contributions and distinctions from existing gradient ascent PGD are:
> - **Different problem formulation.** We optimize perturbations in the *3D Gaussian space*, while enforcing *strict constraints in the multi-view 2D rendering space* (i.e., Eq. (3) in our manuscript). This creates a fundamental mismatch between the optimization space and the constraint space, which, to the best of our knowledge, has not been explicitly addressed in prior 2D editing guard and 3DGS copyright protection works.
>
> - **Different optimization structure (L-PGD).** To solve this cross-space problem, we propose a novel two-stage alternating optimization:
>   - **Gradient truncation in rendering space**, which enforces *hard perceptual bounds* in the intermediate rendering space for strict invisibility constraints,
>   - **Image-to-Gaussian fitting**, which propagates these constrained perturbations back into the 3D Gaussian parameters.
>
>   This *lift-and-fit* mechanism is different from standard PGD: rather than projecting in a fixed space, we iteratively bridge two different spaces (2D $\leftrightarrow$ 3D) to ensure both perceptual control and cross-view consistency.
>
> Importantly, this design is necessary for enabling adversarial protection in 3DGS:
> - Direct 2D PGD fails due to view inconsistency,
> - Direct 3D optimization fails due to lack of perceptual control,
> - Our cross-space formulation + L-PGD can simultaneously achieve view-consistency, strict invisibility, and protection strength to editing models.
>
> We will further clarify these distinctions and strengthened the discussion of our technical contributions in the revised manuscript.

---

> > ### Author Rebuttal · Reviewer_FdAC · 2026-04-03
> >
> > Thank you for the rebuttal and my concerns have been largely resolved.

---

> > > ### Author Response · Authors · 2026-04-07
> > >
> > > Dear Reviewer `FdAC`,
> > >
> > > Thank you for your positive feedback and for your recognition of our work. We are very glad that our rebuttal has resolved your concerns.
> > >
> > > We sincerely appreciate your support and high evaluation, which are a strong encouragement to our work. We will further improve the manuscript based on your suggestions in the revision.
> > >
> > > Best regards,
> > > Submission11967 Authors

---

### Official Review · Reviewer_MD1E · 2026-03-14

**Soundness:** 3
**Presentation:** 3
**Significance:** 3
**Originality:** 2
**Overall Recommendation:** 5
**Confidence:** 4

**Summary:**

This paper proposes a novel pipeline to add protective watermark to Gaussian Splatting asset, so that the protected GS asset will not be easily edited using diffusion model. To ensure invisibility, the author propose lifted PGD, which lift the adversarial optimization on image space to the raw GS space, ensuring the invisibility from different views. Experimental results show that AdaLift can protect the GS asset well, and can generalize to novel views.

**Compliance With Llm Reviewing Policy:**

Affirmed.

**Key Questions For Authors:**

none

**Strengths And Weaknesses:**

Strengths:
- The motivation is good, this paper is working on a important problem.
- The paper is well-written and easy to follow. The tables are detailed and the figures are well-drawn.
- The proposed method is not very new but very solid, it solves an important problem in the field of protecting  GS asset.
- Extensive experiments have beend done and results are solid.

Weaknesses:
- For purification, it was not tested on strong purification methods e.g. PDM-Pure [a]
- For the algorithm, why line 9-11 can ensure that optimized GS render right images under views other than v_t. Will this optimization of GS also affect the results of other views?
- Why you call it Gradient truncation? What's the difference of it from Projected Gradient Descent (PGD)?


[a] Pixel is a Barrier: Diffusion Models Are More Adversarially Robust Than We Think

---

> ### Author Rebuttal · Authors · 2026-03-31
>
> Dear Reviewer `MD1E`,
>
> Thank you for your constructive feedback. We address your concerns below.
>
> > **W1: Robustness against PDM-Pure purification**
>
> **R1:** We thank the reviewer for this suggestion. We conducted additional experiments with PDM-Pure [A1]. From the results in **Tab. A1** (**AdLift\*-VU** and **AdLift\*-VU + PDM-Pure**), while strong purification partially mitigates adversarial perturbations, AdLift still maintains a degradation of editing quality compared to unprotected assets.
>
> Besides, we additionally demonstrate the *compatibility* of our AdLift framework with purification-robust adversarial methods. We combine AdLift with BlurGuard [A2] to leverage low-pass filter to enhance robustness against purifications. As shown in **Tab. A1** (**AdLift-VU(R)** and **AdLift-VU(R) + PDM-Pure**), the combination significantly improves robustness against purification on both the training and novel views. This highlights a key advantage of our AdLift framework: it is not tied to a specific 2D adversarial strategy and can benefit from future advances in purification-robust 2D methods for 3DGS representation.
>
> We will include this discussion in the revised manuscript and further elaborate on its implications for robustness.
>
> **Table A1:** Robustness against purifications.
> | Views |Method|CLIP_d($\downarrow$)|CLIP_s($\downarrow$)|FID ($\uparrow$)|F_1/8 ($\downarrow$)|F_8 ($\downarrow$)|
> |:-:|:-:|:-:|:-:|:-:|:-:|:-:|
> | Train|Unprotected|0.1937|0.9080|23.7996|0.9817|0.9795|
> ||AdLift*-VU|0.1781|0.7900|84.2180|0.8030|0.7749|
> ||AdLift*-VU + PDM-Pure|0.1831|0.8466|47.2951|0.9545|0.9336|
> ||AdLift-VU(R)|0.1776|0.8137|72.7606|0.7778|0.8636|
> ||AdLift-VU(R) + PDM-Pure|0.1779|0.8115|70.8778|0.8888|0.8675|
> | Novel|Unprotected|0.1938|0.9053|51.7357|0.9905|0.9913|
> ||AdLift*-VU|0.1791|0.8037|102.6782|0.8337|0.8794|
> ||AdLift*-VU + PDM-Pure|0.1849|0.8475|75.7447|0.9577|0.9205|
> ||AdLift-VU(R)|0.1756|0.8104|100.5148|0.7548|0.8711|
> ||AdLift-VU(R) + PDM-Pure|0.1792|0.8082|99.7782|0.8448|0.8819|
>
>
> > **W2: The effect between viewpoints**
>
> **R2:** Thanks for the question. **Intuitiely**, the optimization is performed *iteratively across all training views* (line 3 of Algorithm 1). While updating Gaussian parameters on a single view (lines 9–11) may temporarily affect the rendering quality of other views, the subsequent iterations on those other views will compensate and re-optimize accordingly. Since all views share the same underlying Gaussian parameters, this cross-view iterative process progressively drives the 3D representation toward a solution that is jointly optimized across all viewpoints. **Empirically**, quantitative results (e.g., invisibility and protection metrics in Tab. 1 in the manuscript and the adversarial loss curves in Fig. 10) and qualitative results (learned perturbations in Fig. 5 and protection visualizations in Fig. 6) consistently demonstrate effective protection on across multiple *training views* and *novel views* (unseen during optimization), confirming that the protection generalizes well beyond the training viewpoints.
>
>
>
> > **W3: Gradient truncation vs. Projected Gradient Descent (PGD)**
>
> **R3:** We are sorry for the confusion. We clarify the distinction as follows:
> - **Standard PGD on 2D images:** Adversarial perturbations are learned directly on pixel grids, where invisibility is enforced by a strict unified pixel-level perturbation budget (like 16/255 for each pixel). The perturbation and the budget live in the same space (pixels), so gradient ascent followed by projection naturally applies.
> - **Our L-PGD on 3DGS:** 3DGS adopts fundamentally different representations (i.e., Gaussian primitives with heterogeneous attributes such as position, covariance, opacity, and SH coefficients [A3]), making it nontrivial to set a uniform perturbation budget. Therefore, we cannot directly apply gradient projection on Gaussian attributes and applying gradient projection as in standard PGD. Instead, L-PGD reformulates the optimization by shifting the constraint enforcement to the rendered image space: it first truncates the adversarial gradient at the image level to produce a bound-consistent surrogate target, and then fits the Gaussian parameters to this target. Therefore, we ensure the imperceptibility and effectiveness of 3D Gaussian-represented adversarial perturbations. We call this "gradient truncation" because the gradients from the adversarial objective undergo truncation and projection at the image level before being propagated to update the Gaussian parameters.
>
>
> ---
>
> [A1] Pixel is a Barrier: Diffusion Models Are More Adversarially Robust Than We Think. NeurIPS 2024 Workshop SafeGenAi \
> [A2] BlurGuard: A Simple Approach for Robustifying Image Protection Against AI-Powered Editing. NeurIPS 2025 \
> [A3] 3D Gaussian Splatting for Real-Time Radiance Field Rendering. ACM Transactions on Graphics

---

> > ### Author Rebuttal · Reviewer_MD1E · 2026-04-03
> >
> > The empirical gurantee of budges do not sound well to me. But overally it is a good paper and should be accepted.

---

> > > ### Author Response · Authors · 2026-04-08
> > >
> > > Dear Reviewer `MD1E`,
> > >
> > > Thank you for your follow-up. We sincerely appreciate your positive assessment and recognition of our work.
> > >
> > > We agree that the current manuscript focuses on empirical validation of the visual quality and budget-preserving behavior, and does not yet provide a formal theoretical analysis of the approximation error introduced by the image-to-Gaussian fitting step. More specifically, the hard constraint is explicitly enforced in the rendered image space, while the subsequent image-to-Gaussian fitting step lifts this constrained target back into the 3D Gaussian representation through iterative optimization. As a result, our current evidence for the overall cross-view budget-preserving behavior is empirical. Our current work mainly focuses on addressing the practically important problem of proactively safeguarding 3DGS assets against instruction-driven editing, while empirically preserving a hard-bound-like perturbation constraint in the 3DGS optimization process. We view establishing formal guarantees for such cross-space optimization as an important future direction, as it could further strengthen the theoretical foundation of our method.
> > >
> > > We appreciate this important point and will revise the manuscript to make this scope clearer, i.e., to distinguish more carefully between the strict constraint imposed in the intermediate image space and the empirical stability observed after lifting into the 3D Gaussian representation. We will include additional discussion in the revision.
> > >
> > > Thank you again for your time and valuable suggestions.
> > >
> > > Best regards,
> > > Submission11967 Authors

---

### Decision · Program_Chairs · 2026-04-30

**Decision:**

Accept (spotlight)

**Comment:**

This paper ultimately received three recommendations of Accept and one Weak Accept. After the rebuttal, Reviewer FdAC and Reviewer 6Qbd marked their concerns as fully resolved and maintained strong support for acceptance (FdAC had already framed the work very positively, including as a potential Oral/Spotlight candidate subject to contribution clarification). Reviewer MD1E remained partially resolved and continued to recommend acceptance and viewed the work as technically solid. Reviewer 82ur initially gave Weak Accept. After the rebuttal, the reviewer marked prior concerns as fully resolved.

Considering the broadly positive consensus, the substantive rebuttal additions, and the fact that remaining gaps are primarily about scope and theory rather than invalidating the empirical contribution, the AC recommends accepting this paper. Congratulations!

When preparing the final version, the authors should weave the rebuttal clarifications into the main paper; clean up notation globally; fully define auxiliary losses used in localized editing variants; and incorporate purification and modern-editor discussions with the added tables/figures and repository artifacts as appropriate.